

# Droplet Clustering in Shallow Cumuli: The Effects of In–Cloud Location and Aerosol Number Concentration

Dillon S. Dodson.[1] and Jennifer D. Small Griswold[1]

[1]Department of Atmospheric Sciences, University of Hawaii, Manoa, Honolulu, HI, USA

**Correspondence:** Jennifer D. Small Griswold (smalljen@hawaii.edu)

**Abstract.** Aerosol–cloud interactions are complex, including albedo and lifetime effects that cause modifications to cloud characteristics. With most cloud–aerosol interactions focused on the previously stated phenomena, there has been no in–situ studies that focus explicitly on how aerosols can affect droplet clustering within clouds. This research therefore aims to gain a better understanding of how droplet clustering within cumulus clouds can be influenced by in–cloud droplet location (cloud edge vs. center) and aerosol number concentration. The pair–correlation function (PCF) is used to identify the magnitude of droplet clustering from data collected onboard the Center for interdisciplinary Remotely–Piloted Aircraft Studies (CIRPAS) Twin Otter aircraft, flown during the 2006 Gulf of Mexico Atmospheric Composition and Climate Study (GoMACCS). Time stamps (at $10^{-4}$ m spatial resolution) of cloud droplet arrival times were measured by the Artium Flight Phase–Doppler Interferometer (PDI). Using four complete days of data with 81 non–precipitating cloud penetrations organized into two flights of low (L1, L2) and high (H1, H2) pollution data shows more clustering near cloud edge as compared to cloud center for all four cases. Low pollution clouds are shown to have enhanced overall clustering, with flight L2 being solely responsible for this enhanced clustering. Analysis suggests cloud age plays a larger role in the clustering amount experienced than the aerosol number concentration, with dissipating clouds showing increased clustering as compared to growing or mature clouds. Results using a single, vertically developed cumulus cloud demonstrate more clustering near cloud top as compared to cloud base.

## 1   Introduction

The physical processes controlling clouds are complex, with two of the largest uncertainties being precipitation formation and aerosol–cloud interactions, both of which can affect cloud lifetime and size. Along with these uncertainties, one of the main problems with cloud microphysical research has been determining the importance of turbulence on extremely small scales in affecting the macroscopic evolution of clouds, along with gathering in–situ data to better understand these properties (Shaw, 2003). These uncertainties and problems lead to clouds being one of the largest inaccuracies when estimating climate sensitivity.

Focusing on precipitation formation, there appears to be a factor of two or more difference between the predicted growth time of precipitation calculated by Jonas (1996) of 80 minutes and the observed growth time of 15 to 20 minutes from radar measurements made by Laird et al. (2000) and Szumowski et al. (1997), as discussed in Lehmann et al. (2009). Cloud droplets of radii less than 10 to 15 $\mu$m grow efficiently through diffusion of water vapor while droplets larger than 30 to 50 $\mu$m grow





efficiently through gravitational collisions (Langmuir, 1948; Kogan, 1993; Pruppacher and Klett, 1997). In general, it is hard to explain the rapid growth of cloud droplets in the size range from 10 to 50 $\mu$m (the so–called size gap), for which neither the condensation nor the gravitational collision–coalescence mechanism is effective. Multiple theories have been proposed to explain the short rain formation time observed, including: turbulent deviations in supersaturation (Vaillancourt et al., 2002);

entrainment mixing (Khain et al., 2000; Lehmann et al., 2009); giant (GN) and ultragiant (UGN) nuclei (with diameters greater than 2 $\mu$m and 10 $\mu$m, respectively (Knight et al., 2002)); and the turbulent enhancement of collision–coalescence due to droplet clustering. Although each of these theories has merit, no one theory has been able to explain the fast formation of rain that is observed (i.e., Srivastava (1989); Vaillancourt et al. (2002); Rosenfeld et al. (2002); Feingold et al. (1999)).

Shifting our focus to aerosols, it is known that the effects of aerosols on clouds can lead to increased cloud lifetime due to the

suppression of rain formation and to higher cloud albedo due to smaller droplets, better known as the cloud lifetime and albedo effects, respectively (Small et al., 2009). Enhanced evaporation from smaller droplet sizes arises from aerosol perturbations, resulting in a stronger horizontal buoyancy gradient and increased entrainment, known as the evaporation–entrainment feedback mechanism (Small et al., 2009), where increased (decreased) entrainment could lead to decreased (increased) cloud lifetimes. This suggests that aerosol perturbations can lead to modifications of the turbulent environment within clouds, specifically at

the entrainment interface. According to Ramaswamy et al. (2001), although increased aerosols are known to casue a negative forcing on the radiative budget when interacting with clouds, the magnitude of that negative forcing has the largest uncertainty as compared to other components considered (greenhouse gases, surface albedo, etc.) when studying factors contributing to cliamte change.

The complexity of clouds can clearly be seen from the discussion above, and a better understanding of said complexity

can be gained by understanding droplet clustering on the smallest scales. Up until the late 1980's, it was mostly accepted that droplet spacing within clouds was statistically homogeneous, or uniformly distributed according to Poisson statistics (Marshak et al., 2005; Rogers and Yau, 1989). Srivastava (1989) argued that in most numerical studies of cloud physics it is assumed that droplet to droplet variability is not important in calculating the growth of an ensemble of droplets. However, this conclusion must be viewed as tentative due to evidence that has been gathered over the past three decades for droplet clustering occurring

specifically at the millimeter and centimeter scales (e.g., Baker (1992); Kostinski and Jameson (1997); Kostinski and Shaw (2001); Larsen (2007); Siebert et al. (2010); Shaw et al. (1998, 2002)). The presumption that droplet spacing is homogeneous has consequences for cloud parameterizations in microphysical models such as the formation of precipitation. For example, the stochastic collection equation, used to describe the growth of droplets via collision–coalescence, assumes that droplets are homogeneously distributed and not preferentially concentrated (Kostinski and Shaw, 2001). An enhanced collision kernel

and collision efficiency also results from droplet clustering (Pinsky et al., 1999). Quantitative results from Pinsky et al. (2008) found that droplet growth can be enhanced by a factor of 2–3 through turbulent collision–coalescence, while Grabowski and Wang (2009) found that growth was enhanced between factors of 1.2 to 4.5 based on different values of turbulent kinetic energy (TKE) dissipation rates.

One of the main difficulties in turbulence and droplet clustering work as explained by Pinsky et al. (2006) is no measure-

ments of the fine turbulent structure in clouds have been carried out. Most laboratory experiments are conducted for Reynolds





number values that are much smaller than those typically found for atmospheric turbulence. Since the structure of a turbulent flow depends on the Reynolds number, it is not clear how the results obtained in laboratory experiments can be extended to atmospheric conditions. The work conducted here therefore becomes important due to the fact that a dataset is provided that gives in–situ cloud droplet spatial data that allows for an analysis of spatial variability, and the fact that droplet spatial variations

can be compared to laboratory measurements to see if they can indeed be extended to atmospheric conditions.

This research focuses on the fundamental investigation of droplet clustering, specifically how it changes within clouds (cloud edge vs. cloud center, as a function of cloud height) and different cloud environments (low vs. high pollution clouds). Specific questions in regards to droplet clustering in shallow, warm continental cumulus clouds to be answered include: (1) Does droplet clustering depend on aerosol number concentration? (2) Does droplet clustering change as a function of location (cloud center

vs. edge)? (3) Does droplet clustering change as a function of cloud height? This information can eventually be used to develop better cloud microphysical parameterizations not only for modeling precipitation, but for modeling the overall role of clouds in radiation models. Section 2 will introduce droplet clustering and the statistical tool used to measure the clustering. Section 3 will discuss data collection and instrumentation along with environmental and flight characteristics. Section 4 will provide results related to the three scientific questions proposed above. Section 5 will lead to a discussion of the results, including a

hypothesis on how clustering is influenced by cloud age, followed by concluding remarks.

## 2   Droplet Clustering and the Pair-Correlation Function (PCF)

### 2.1   Droplet Clustering

It has been proposed in multiple studies (i.e., Shaw et al. (1998); Eaton and Fessler (1994); Sundaram and Collins (1997)) that droplet clustering (also known as preferential concentration or inertial clustering) can be understood as the result of particles

being centrifuged out of regions of high fluid vorticity (where vorticity is a measure of local rotation in fluid flow) and thus preferentially concentrating into regions of high strain or low fluid vorticity as a consequence of their inertia. Sundaram and Collins (1997) have shown that the most responsible scale for preferential concentration is the Kolmogorov scale (mm to cm, depending on the rate of turbulent dissipation). This is partially supported by the fact that vorticity plays a key role in concentrating particles, and vorticity is predominantly concentrated in the smallest eddies (Tennekes and Lumley, 1972). This

is due to turbulence being dissipative. The dissipation of energy is most effective at small scales, which together with the fact that energy is mainly provided at large scales implies that energy is transferred from larger to smaller scales (energy cascade). Fluid inertia dominates over viscosity at larger spatial scales, while viscosity dominates at smaller scales (dissipation range). Preferential concentration is enhanced between the dissipation and inertial ranges because it is at these scales that turbulent vorticity is dominant (Wang and Maxey, 1993). Dissipative structures that appear within turbulence can be referred to as vortex

tubes that infuse isotropic turbulence, where turbulence is said to be isotropic if rotation and buoyancy are not important and can be neglected, and there is no mean flow. Vortex tubes are relevant to the present discussion because they are thought to be responsible for ejecting particles into high strain regions of the flow (Shaw et al., 1998).

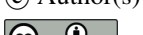



Droplet clustering, for small Stokes numbers, can be understood mathematically by the equation:

$$\frac{\partial U_i}{\partial x_i} = -\tau_d(s_{ij}{}^2 - \Omega_{ij}{}^2) \tag{1}$$

where $U_i$ is droplet velocity, $s_{ij}$ is the strain rate tensor, $\Omega_{ij}$ is the rotation tensor or vorticity tensor, and $\tau_d$ is the droplet response time (Maxey, 1987). The conclusion can be made from Equation 1 that the droplet velocity field is divergent in regions of high vorticity ($\Omega_{ij}$) and convergent in regions where strain ($s_{ij}$) dominates (Maxey, 1987; Shaw, 2003). This makes sense in an analysis of the full derivation, where the divergence of the droplet velocity is taken to obtain Equation 1, keeping in mind that divergence (convergence) results when the divergence equation is positive (negative).

The particle response time ($\tau_d$) is not the only value that determines whether or not particles will tend to cluster. The ratio of $\tau_d$ to the relevant timescale of fluid accelerations is also of importance. Since clustering is associated with vorticity, and the vorticity spectrum peaks at small scales, the Kolmogorov timescale $\tau_k$ is the relevant fluid time scale, where

$$S_t = \frac{\tau_d}{\tau_K} = \frac{\rho_w d^2 \varepsilon^{\frac{1}{2}}}{18 \rho_a \nu^{\frac{3}{2}}} \tag{2}$$

is the Stokes number (Vaillancourt et al., 2002), with $\rho_a$ and $\rho_w$ representing the density of air and liquid droplets, respectively, $d$ the droplet diameter, $\nu$ the fluid kinematic viscosity, and $\varepsilon$ the turbulent energy dissipation rate. The Stokes number characterizes a particles inertial response to the flow. Particles with $S_t \gg 1$ react very slowly to the changes in the flow due to large particle response times while particles with $S_t \ll 1$ follow the flow exactly. Droplet clustering is related to $S_t$ by the power law:

$$\eta(r) \propto \left(\frac{r}{r_K}\right)^{-f(S_t)} \tag{3}$$

where $\eta(r)$ is the spatial pair–correlation function (used to measure the amount of clustering, see Section 2.2), $r_K$ is the Kolmogorov length scale, and $f(S_t) > 0$ increases monotonically with $S_t$ for $S_t < 1$ (Saw et al., 2008). This implies that clustering increases for increasing $S_t$ (for values between zero and one). Hogan and Cuzzi (2001) and Wood et al. (2005) also concluded that clustering is at a maximum for Stokes numbers near one. This suggests that clustering depends on the droplet size and the turbulent dissipation rate. It is known that the typical range of Stokes number in clouds is $S_t \ll 1$ to $S_t < 1$ (Vaillancourt et al., 2002; Fouxon et al., 2015; Moghadaripour et al., 2017), indicating that for the range of Stokes numbers that occur in clouds, clustering increases as the droplet size increases or as the turbulent dissipation rate increases.

It is important to note that how droplet clustering is related to turbulence will be discussed, but no actual turbulence parameters will be measured due to data availability, leaving for the chance of future work to expand on the results presented here. For example, one would expect that turbulence should be enhanced near cloud top and edge where the entrainment of non–turbulent air is mixed with turbulent cloudy air (Siebert et al., 2006). Shaw (2003) also states that one of the main sources of TKE in clouds is shear evaporative cooling due to the entrainment of dry air at cloud edge and top. Direct measurements have shown that the mean TKE dissipation rate peaks near cloud top at its edge (MacPherson and Isaac, 1977; Gerber et al., 2008). It can therefore by hypothesized that droplet clustering will be enhanced at cloud edge (top) as compared to cloud center (bottom) due to increased Stokes numbers. On the contrary, an increase in aerosol number concentration will lead to a decrease





in droplet size (Small et al., 2009; Rosenfeld et al., 2008), a decrease in the Stokes number, and a decrease is droplet clustering. Note that while clustering may well be enhanced at cloud edge or top, the reason for this enhanced clustering is made based on previous work and is speculative due to the fact that TKE is not directly measured here. This work sets out to simply determine how the spatial variability of droplet clustering changes with in–cloud location and aerosol number concentration, leaving the

direct relationship between in–cloud turbulence and clustering for future work.

It is important to state that preferential concentration and the physical processes leading to it at the millimeter and centimeter scales is different than the inhomogeneity in droplet concentration that occurs at larger scales on the order of several meters. Larger scale inhomogeneity is caused by the inhomogeneity of turbulence (fluctuations of vertical velocity) and cloud condensation nuclei (Pinsky and Khain, 2002, 2003), and the entrainment of sub–saturated air containing few or no droplets (Krueger

et al., 1997). The inhomogeneity of the droplet population at larger scales will be accounted for, as will be discussed in Section 2.3.

## 2.2   Pair–Correlation Function

There are multiple tools that can be used to measure droplet clustering using a time series of droplet detection times, but the 1–D temporal pair–correlation function (PCF) will be used throughout this paper due to the advantages of the PCF outlined in Shaw

et al. (2002); Shaw (2003); Larsen (2006, 2012); Baker and Lawson (2010). The PCF can be introduced as a scale–localized deviation from a stationary Poisson distribution, where the PCF is given by:

$$\eta(t) = \frac{p(t_o + t|t_o)}{\lambda} - 1 \qquad (4)$$

from Larsen (2012), where $\eta(t)$ is the PCF, $p(t_o + t|t_o)$ represents: given a particle detected at some time $t_o$, what is the probability of finding another particle in the time lag $t_o + t$. The mean number of droplets per time bin is given by $\lambda$. Calculating

$p(t_o + t|t_o)$ can become simplified by using:

$$\eta(t) = -1 + \frac{1}{\lambda} \sum_{k=1}^{\infty} f_k(t) \qquad (5)$$

where $f_k(t)$ is the probability distribution function that the $k^{th}$ particle posterior to a particle at $t_o$ (the $k^{th}$ nearest neighbor) is located at $t_o + t$, where it is assumed that co–located particles are impossible. Each of the $f_k(t)$ can be estimated from the observed inter–arrival distributions (time between droplet arrival), thus allowing a computationally simple way to compute the

PCF from particle arrival times. For more information and a derivation of the $k^{th}$ nearest–neighbor function see Picinbono and Bendjaballah (2005).

The main advantage of the PCF is the fact that it is scale localized. The PCF depends only on the presence or absence of particles separated by $t$ in time (Larsen, 2012). Physically, when $\eta(t) > 0$ there is an enhanced probability of finding a particle in the time frame $t$. The range of the PCF is $(-1, \infty)$ with $\eta(t) = 0$ representing perfect randomness and $\eta(t) = 3$, for example,

resulting in a factor of 4 enhancement of finding another droplet time $t$ away, as discussed in Kostinski and Shaw (2001).

Although the PCF is a great statistical tool for detecting droplet clustering, its physical sense is not so profound. Along with that, particle positions have been determined in 1–D space, although 3–D space would provide more information on





the overall local concentration enhancement. Both issues can be accounted for from Holtzer and Collins (1997); Sundaram and Collins (1997) and Zhou et al. (2001). It should also be noted that the PCF is referred to as the radial distribution function ($g(r) = \eta(r) + 1$, where $r$ represents distance as compared to time) in these papers (Shaw, 2003). Sundaram and Collins (1997) showed that preferential concentration can be accounted for in the Saffman–Turner collision kernel (Saffman and Turner, 1956)

by a factor equal to $\eta(t) + 1$. This has been shown to be true for both monodisperse ($g_{11}(r)$) and bidisperse ($g_{12}(r)$) systems (Zhou et al., 2001), where $g_{11}(r)$ and $g_{12}(r)$ represents the radial distribution function (RDF) for each system, respectively. The droplets in this paper would be considered bidisperse as droplets of different sizes were used to caluclate the PCF. To infer information about 3–D clustering, it has been shown in Holtzer and Collins (1997) that the 1–D PCF is equivalent to integrals of the 3–D PCF (Equation 2.5 from Holtzer and Collins (1997)). Although measuring the actual physical impact that the PCF

has on collision–coalescence is beyond the scope of this paper, it is important to understand how the results obtained here can be used to better understand the physical processes that occur within clouds.

### 2.3 Calculating the PCF

The PCF is reliant on the fact that the underlying dataset must be statistically stationary/homogeneous (the mean and variance of the number of counts are assumed to be constant over the analyzed time interval). This is due to the fact that the PCF may

deviate from zero not from a statistical correlation of clustering at smaller scales, but from inhomogeneity at larger scales (Larsen (2012) Appendix B). This requirement can be met when sampling horizontally homogeneous clouds such as stratus. For more turbulent cumulus clouds were stronger entrainment and mixing processes occur, this condition is usually only fulfilled for subsections of the cloud.

The PCF was calculated three times for each cloud penetration (120 m section) at cloud edge (cloud entry and exit) and cloud

center. One–hundred and twenty meters represents a two second interval of data, which provided enough cloud droplets to run the PCF while not completely ignoring data stationarity. However, due to the data being slightly non–stationary nonetheless, each PCF calculation was normalized so it decayed to zero. The justification for PCF normalization comes from Shaw et al. (2002), where it was shown that the PCF shape is similar when comparing the PCF of an entire cloud penetration vs. just cloud center. The two PCF curves were separated by a vertical shift due to droplet concentration fluctuations occurring on

scales larger than $t$ from non–stationarity. Qualitatively however, the curves were identical. The small–scale clustering was shown to not be influenced by large scale fluctuations, making normalization possible (see Figure 3 in Shaw et al. (2002) for more information). Note that the PCF is a useful quantitative and qualitative tool, but anything quantitative must be carefully evaluated if it comes from data that is either nonstationary or not known to be stationary, again, stating the importance that this paper is only looking at how clustering changes with cloud location and with aerosol number concentration, and not drawing

any quantitative results from the clustering on processes such as collision–coalescence.

To calculate the $k^{th}$ nearest neighbor, a maximum time interval (t–max) and time bin ($dt$) had to be selected. Careful consideration had to be given. Set $dt$ too small, the PCF will be too noisy. Set $dt$ too large, you end up doing unnecessary scale averaging which results in a poor estimate of the PCF. Typically, t–max is an order of magnitude or so above the mean inter–arrival time (MIT, mean time between each droplet within the data) of the particles and sets the maximum temporal lag. A



dt of 0.0003 seconds and a t–max of 0.2 seconds were selected for all PCF calculations throughout this paper. This results in a vector ranging from 0.000 to 0.2 by 0.0003, giving the temporal lag (x–axis) for each PCF measurement. The PCF is calculated by binning the inter–arrival times of the droplets into the vector sequence seen above. An inter–arrival time is first determined between every subsequent droplet, binned and summed (the sum for each inter–arrival time per bin). An inter–arrival time is
then determined for every other droplet, every third droplet, every fourth droplet, and so on. The inter–arrival times are binned and added to the previously summed binned inter–arrival times up until the minimum inter–arrival time in the data is no longer less than t–max. The total summed binned data is then used to calculate the PCF from Equation 4.

Figure 1 gives a visualization and description of the PCF clustering signature, giving values for temporal and spatial lag on the x–axis. Note that the results will present the PCF in terms of spatial lag (where spatial lag was estimated using the mean
aircraft velocity) for the simplicity of being able to more easily comprehend spatial lag over temporal lag. The real data (top left) shows a peak at smaller spatial scales and a steady decrease to zero at larger spatial scales while the Poisson data (top right) shows the PCF varying around zero, indicating no clustering at any scale. The mean of the two PCFs are 0.057 and $1.94 \cdot 10^{-5}$ for the real and simulated data, respectively. The mean was calculated by taking the first 8 PCF values (covering a spatial scale up to 13 cm), since it is at smaller spatial scales in terms of analyzing droplet clustering that we are concerned
with. This displays that real cloud droplets have a greater amount of clustering as compared to droplets that have a perfectly random orientation. From a visual examination of the raw droplets (bottom panels), the real data is preferentially concentrated or patchy, whereas the Poisson point data is perfectly homogeneous.

## 3   Data Collection and Characteristics

The Gulf of Mexico Atmospheric Composition and Climate Study (GoMACCS) was conducted jointly with the 2006 Texas Air
Quality Study (TexAQS) during August and September of 2006 as a combined climate change and air quality intensive field campaign. The Center for Interdisciplinary Remotely–Piloted Aircraft Studies (CIRPAS) Twin Otter (flight speed of about 60 m s$^{-1}$) performed 22 research flights to explore aerosol–cloud relationships over the Houston and northwestern Gulf of Mexico regions (Lu et al., 2008). Among the 22 research flights, 14 intensive cloud measurements were carried out (where the clouds were all continental warm cumulus subjected to various levels of anthropogenic influence), including one flight in which an
isolated cumulus cloud of sufficient size and lifetime existed to allow detailed sampling at different altitudes. The other 13 cases involved scattered cumuli that were sampled in such a manner as to provide statistical properties over the cloud field (Lu et al., 2008), with each cloud being traversed through once, with no one cloud being measured multiple times.

Table 1 shows each flight conducted during GoMACCS, with the corresponding Research Flight (RF) number, date, number of clouds in the flight after filtering (including clouds that are only $> 300$ m in length and non–precipitating), the aerosol
number concentration ($N_a$, measured by the condensation particle counter (CPC)), and the aerosol number concentration for accumulation mode particles ($N_{acc}$, measured by the passive cavity aerosol spectrometer probe (PCASP)), which includes aerosols that are only in the size range of 0.1 $\mu$m $<$ particle size $<$ 2.5 $\mu$m. The Phase–Doppler interferometer (PDI, see Chuang et al. (2008)) was used to collect droplet velocity, size, and measurement time. It was found that the droplet arrival





time can accurately be measured to $< 3.5 \ \mu$m from Saw (2008), resulting in accurately mapping droplets down to $2.1 \cdot 10^{-4}$ m (assuming average aircraft speed). Note that there is no dead time in PDI measurements. For more information on each of the flights and the instrument payload, see Lu et al. (2008).

Following the methods in Small et al. (2013), two low (L1, L2) and high (H1, H2) pollution flights were selected out of the 22 research flights that occurred. The two least and most polluted flights were selected which had satisfactory cloud sampling for analysis of how aerosol number concentration effects droplet clustering. A Case Flight (Flight 16) was selected where an isolated cumulus cloud was sampled at different altitudes for analysis of droplet clustering as a function of cloud height. Table 2 shows variables highlighting different cloud and environmental conditions within each flight. Note that the environmental lapse rate and relative humidity (RH) in Table 2 was calculated from data collected from out of cloud spirals, where the average RH was computed for the vertical range of cloud measurments for the respected flight. Table 3 gives a summary of average values for low and high pollution cases for select properties from Table 2.

Figures 2 and 3 show the flight altitude as a function of time with droplet counts per second overlaid (top panels) and the flight paths (bottom panels) for low and high polluted clouds, respectively. Note that the flight path for the Case Flight is not shown here. The average droplet counts (clouds) encountered per second (flight) for L1 and L2 were 660 (18) and 1016 (13), respectively. Whereas for H1 and H2, the average counts (clouds) encountered per second (flight) were 958 (29) and 3300 (21), respectively. Low pollution clouds were sampled to the North of Houston (upwind) and high pollution clouds were sampled to the Southwest (H1) and West (H2) of Houston (downwind), as confirmed using archived wind data from the NOAA National Center for Environmental Information and HYSPLIT trajectories (not shown here) from the Air Resources Laboratory (Stein et al., 2015).

It can be calculated from analyzing Table 3 that the high pollution clouds had roughly 2.5 times more aerosols per cubic centimeter than the low pollution clouds. The difference in aerosol number concentration between the low and high pollution clouds produce clouds that are statistically different from one another. Figure 4 shows cloud droplet diameter in microns ($\mu$m) on the x–axis with aerosol number concentration (cm$^{-3}$) on the y–axis, with low pollution data in green and high pollution data in gold. Density curves are given to show how the data is distributed for the respected axis. The p–value (used to determine statistical significance between two datasets, where p–value $< 0.05$ is considered significant, see Wilks (2011)) between low and high pollution cloud droplet size is $3.99 \cdot 10^{-10}$ (average droplet diameter is 13.4 $\mu$m (10.7 $\mu$m) for low (high) pollution clouds). The linear best fit trend lines show that droplet size decreases with increasing aerosol number concentration, with R–squared values (the proportion of the variance in droplet size that is predictable from the aerosol number concentration) of 0.24 and 0.07 for low and high pollution, respectively. The p–value for the aerosol number concentration is $< 2.22 \cdot 10^{-16}$. Both properties of the droplet population have p–values less than 0.05, making the difference in droplet size and aerosol number concentration significant for the two populations of data. Having two statistically different data populations is ideal for comparing PCF values for low and high pollution clouds. If clustering does not change between the two, then an argument cannot be made for the statistical similarities in the data sets as a possible reason. Note that all p–values in this paper were calculated using the Wilcoxon–rank–sum–test, which is used to determine a statistical difference in the medians of two datasets that have different populations (Wilks (2011)).



## 4   Results

### 4.1   Edge, Center, and Cloud Top Clustering

PCF functions for L1, L2, H1, and H2 are given in Figure 5, moving from top left to bottom right, respectively, with blue (red) representing entrainment zone (cloud center) data. The two envelopes represent the $85^{th}$ and $15^{th}$ percent quantile of the data.

The center lines in each envelope represent the entrainment and center mean clustering, with bold mean lines representing data that is statistically significant (p–value less than 0.05) and thin mean lines representing data that is statistically similar. The PCF function for both the center and entrainment zone is at a maximum for lower spatial scales ($< 30$ cm) and decreases towards zero at larger spatial scales for all four flights. The main takeaway from Figure 5 is the more dominant clustering signature at smaller spatial scales and the enhanced clustering for the entrainment as compared to the center zones for all four flights. Mean

PCF and quantile values for entrainment and center data can be found in Table 4. The percent of statistically significant data (for the first 21 PCF values, spatial lag $\leq 36$ cm) and the corresponding p–values can be found in Table 5. Note to calculate the p–value, every PCF curve generated for the respective plot was grouped. A p–value was then generated for each spatial-lag on the x–axis by calculating the Wilcoxon–rank-sum–test between the two sets of data for the specific x–axis location.

From analyzing Figure 5 and the corresponding tables, L1, H1 and H2 show clustering characteristics which are comparable

to one another, including: (1) the mean PCF value for the entrainment data is always greater than the mean PCF value for the center data. (2) The $15^{th}$ percent quantile value for the center data is always smaller than the $15^{th}$ percent quantile value for the entrainment data. (3) The $85^{th}$ percent quantile value for the entrainment data is always larger than the $85^{th}$ percent quantile value for the center data. (4): There is a statistical significance between the clustering occurring between the entrainment and center zones of the clouds. Note that the statistical significance in the clustering between the two zones breaks down at larger

spatial scales (this is very apparent in the H1 case). This is expected, since the two datasets converge to zero as the clustering disappears and droplet spacing shifts from non–homogeneous to homogeneous at larger spatial scales. (5) The mean PCF values for entrainment data are very similar, ranging from 0.50 to 0.54, whereas the range for the center data is between 0.18 to 0.30 (Table 4).

From analyzing L2 (bottom left panel) and the corresponding tables, there are significant differences from the other 3

cases. Although the mean entrainment clustering is enhanced as compared to the center zone, the difference is not statistically significant, with zero percent of the data having a p–value below 0.05 (average p–value of 0.30). Another difference is the range of the $85^{th}$ percent quantiles, with the two quantile values being virtually similar at 1.79 and 1.81 for center and entrainment zone data, respectively. Note that the mean clustering amount (both center and entrainment) is enhanced as compared to the other three cases.

It is clear that there is enhanced clustering in the entrainment zone as compared to the center zone, but one needs to understand how to define if the overall clustering (both entrainment and center) is significant. This is done by analyzing the range that the PCF can take on due to the random nature of the data. If the physical clustering measured falls outside of this range, then the conclusion can be made that the clustering being viewed is indeed real and not perfectly homogeneous. This test was performed on each of the four cases, following the methods outlined in Larsen and Kostinski (2005). For the data, 1000 Poisson





simulations were produced (as is seen in the top right of Figure 1, showing a single Poisson simulation) using the same time duration and droplet count as the original data. These Poisson simulations then form an envelope of PCF values (using the maximum and minimum values from the 1000 simulations) one would consider homogeneous. PCF values that lie within the Poissonian simulation envelope were recorded by using the average PCF value and were labeled non–significant. Table 6 shows

the percentage of PCF values for each flight, and each location (entrainment and center), that were considered non–significant and significant. It can be seen that except for the L2 case (which does not have a statistical difference between entrainment and center clustering) that the center clustering contains a higher percentage of PCF values that are non–significant as compared to the entrainment data.

Figure 6 shows how the PCF and other environmental properties (cloud droplet number concentrtion (cm$^{-3}$), liquid water

content (g m$^{-3}$), RH (%), and vertical velocity (m s$^{-1}$)) vary with normalized cloud height. Variable quantities for each normalized cloud height can be found in Table 7. The liquid water content (LWC) increases from cloud base (0.073 g m$^{-3}$) to a normalized cloud height of 0.7 (0.98 g m$^{-3}$) before decreasing to 0.23 g m$^{-3}$ at cloud top. Accompanied by the decrease in LWC is a sharp decrease in the RH from 99.74 % to 64.17 % between normalized cloud heights of 0.7 and 0.9, before increasing again at cloud top to 95.7 %. As both the LWC and RH decrease, the PCF has a sharp increase from 0.26 to 1.81

between normalized cloud heights of 0.8 and 1.0, indicating enhanced clustering at cloud top. Panel (a) gives the cloud drop size distribution for each normalized cloud height. The median drop size increases from 11.88 $\mu$m at cloud base to 17.65 $\mu$m at cloud top. In comparing the median drop size to the mean PCF value for each normalized height, the R–squared value is 0.07, indicating no correlation between the PCF and median droplet size. Panel (c) shows that the vertical velocity is negative in the upper portion of the cloud, while an updraft is present in the lower 50 % of the cloud. Table 8 gives the p–value between

every normalized cloud height for the PCF. It is important to note that PCF values between normalized cloud heights of 0.8 to 0.9 are statistically significant, making clustering that is present from a normalized cloud height of 0.9 to cloud top statistically significant from the clustering that is occurring in lower cloud layers.

## 4.2 Low vs. High Pollution Clustering

Figure 7, Panels (1a) and (1b) gives the same information as in Figure 5, except for the PCF values for total low (average of

L1 and L2) and total high pollution (average of H1 and H2), respectively. The characteristics of the two clustering signatures are similar to that of Figure 5. The average PCF values for low and high pollution entrainment and center data can be found in Table 4, along with the 15$^{th}$ and 85$^{th}$ percent quantile values. Table 4 reveals that the mean PCF values (for both entrainment and center data) for the low pollution case are larger than the corresponding mean PCF values for the high pollution case. As can be seen in Table 5, 100 percent of the first 21 spatial lags are statistically significant for both average low and high pollution

cases between the entrainment and center data.

The larger average clustering amount for the low pollution clouds can be seen well in Panel (2a), which shows low pollution data in green and high pollution data in gold. The boundaries of each green and gold envelopes are created by the average center (bottom of each envelope) and entrainment (top of each envelope) clustering. Low pollution clouds are clearly offset to a higher clustering amount for both average center and entrainment clustering. Panel (2b) shows the overall average of all the





PCF values for low and high pollution clouds. The overall average PCF value for low pollution clouds (average of entrainment and center clustering for both L1 and L2) is 0.57, while the overall average PCF value for high pollution clouds is 0.44. Although it appears that low pollution clouds experience more clustering as compared to high pollution clouds, the difference is statistically similar. The average p–value is 0.10 for the first 21 time lags with zero percent of the data being statistically significant.

Although it appears that low pollution clouds have a non–statistically significant higher amount of clustering than high pollution clouds, further analysis shows that the higher amount of clustering in the low pollution case is due entirely to the L2 flight. Figure 8 gives the same information as Panels (2a) and (2b) in Figure 7, except for the individual flights (L1, L2, H1, H2) are shown. From analyzing Panel (a), one can see the average center and entrainment clustering for L2 (light green envelope) is beyond the range of the other three flights. The total average PCF for the clouds in L1, L2, H1, and H2 is shown in Panel (b). L2 has an average PCF value of 0.77, which is roughly twice the average PCF value (and statistically significant, see Table 9) of the other three flights, where L1 (dark green), H1 (dark gold), and H2 (light gold) have average PCF values of 0.42, 0.44, and 0.43, respectively. The question of whether clustering depends on aerosol number concentration cannot confidently be answered. Although Figure 7 shows that low pollution clouds have a larger amount of clustering, statistically speaking the clustering between low and high pollution clouds is the same. Further analysis shows that L1, H1, and H2 all have statistically similar clustering values (see Table 9) with average clustering amounts that are almost identical. Flight L2 has statistically significant clustering as compared to the other three cases, and is solely responsible for causing the low pollution clouds to have a higher average PCF value than that of the high pollution clouds.

## 5 Discussion

### 5.1 Cloud Lifetime Theory and Clustering in L2

An explanation for the statistically different clustering in L2 as compared to the other three cases could be cloud age. A study by Schmeissner et al. (2015) found that dissipating clouds have five main characteristics, including: a negative buoyancy (m s$^{-2}$) and vertical velocity, lower LWC and cloud droplet number concentrations (CDNC) as compared to actively growing clouds, and a larger RH shell around the cumulus cloud. Cooper and Lawson (1984) also found that the LWC decreases due to entrainment as cumulus clouds deteriorate.

Figure 9 shows box plots of vertical velocity (a), LWC (b), cloud width (c), CDNC (d), buoyancy (e), and RH (f) on the y–axis and L1, L2, H1, and H2 represented in that order on the x–axis. Red median lines represent datasets that are statistically different when compared to L2. Note that except for cloud width, each variable is represented from 1 Hz data collected during in–cloud sampling. From analyzing Figure 9 (exact median values for variables can be found in Table 10), L2 has the lowest median vertical velocity, LWC, cloud width, and CDNC, with L2 being statistically significant (p–values found in Table 11) when compared to the other three flights for each of the variables. The fact that L2 has the lowest median vertical velocity reflects the fact that clouds in L2 are either dominated by downdrafts, or have weak updrafts, making growth unlikely. A low LWC signifies that entrainment of dry air has been occurring, resulting in the evaporation of liquid water droplets, reducing the



LWC and the CDNC (Pruppacher and Klett, 1997). Although Schmeissner et al. (2015) does not discuss cloud width, clouds that are dissipating would be expected to have a smaller horizontal extent due to entrainment of dry air leading to evaporation of cloud edge droplets as compared to mature clouds.

Panel (f) gives a box plot of in–cloud RH, with L2 having the lowest (by 0.1 percent) in–cloud RH, while being statistically

similar to that of L1. The red dots represent the median out–of–cloud RH (100 m before and after cloud edge). L2 is the only flight where the RH increases out of the cloud. According to Schmeissner et al. (2015), the width of the humid shell around clouds is larger for dissolving clouds as compared to actively growing clouds. The fact that the RH is larger, on average, outside of the clouds in L2 as compared to inside of the clouds could be a sign of a large humid shell that is surrounding the individual clouds. The humid shell results from entrainment of dry air into the cloud, while moist air is detrained out of the cloud into the

cloud free environment, resulting in a lower (larger) RH inside (outside) the cloud. More evidence for the large humid shell can be gathered from the vertical profiles of environmental RH reported in Table 2, where the average RH (measured out–of–cloud) for the vertical range of cloud measurements was 105.2 % for L2, while for the other flights the RH was considerably lower.

Panel (e) shows the in–cloud buoyancy, which was calculated by taking the in–cloud and out–of–cloud (100 m before and after cloud edge) virtual potential temperatures. L2 has the largest median buoyancy and is statistically significant as compared

to the other three flights. The clouds in the L2 flight have five out of the six characteristics for decaying clouds, including (1) lowest vertical velocity; (2) lowest LWC; (3) lowest CDNC; (4) lowest cloud width; (5) largest humid shell. The evidence points to the clouds in L2 to be decaying on average, and therefore to be more turbulent as dry air is mixed into the clouds causing dissipation. The statistically similar values between center and entrainment clustering for L2 adds to the cloud lifetime theory, as it is not only the entrainment zone that is experiencing mixing, but the entire horizontal extent of the clouds (both entrainment

and center zones) that are experiencing mixing and dissipation. However, one would expect the buoyancy of dissipating clouds to be negatively buoyant, not positively buoyant as is shown. Although cloud age is a good theory in describing the higher clustering amounts measured in the L2 flight, the data presented does not offer a conclusive resolution.

Adding to the discussion, the larger the age of a cloud the higher the typical Stokes number will be due to larger droplets (from a longer droplet growth duration), implying larger clustering values. For the clouds measured in this study, the median

droplet size was 15.01 (L1), 11.45 (L2), 11.19 (H1), and 10.92 $\mu$m (H2). Although the droplet size of each droplet population is statistically significant form one another (all p–values are $\leq 2.32\cdot10^{-2}$), the resulting Stokes numbers for each flight (assuming a constant TKE dissipation rate of 100 cm $^2$ s$^{-3}$ are 0.018, 0.010, 0.0097, and 0.0093 for L1, L2, H1, and H2, respectively. These Stokev values are all very similiar to one another, suggesting that the significant difference in clustering seen in L2 is due to increased turublence from mixing, and not a difference in droplet size.

Other possible explanations for the increased clustering in L2 could be due to flight path or the atmospheric environment for a given flight. The flight path through the cumuli for L2 could have favored cloud edge or cloud top instead of true cloud center. Favoring cloud edge would result in measuring areas of cloud that favor a higher amount of clustering (as displayed in Figure 5), and could result in the overall larger amount of average clustering experienced. Measuring just cloud edge would result in: a lower vertical velocity, LWC, and CDNC (due to evaporation from entrainment), and a shorter cloud width from



not traversing the maximum diameter of the cloud. However, just as with the cloud lifetime theory, the buoyancy is expected to be negative, not positive, in the entrainment zone of the cloud due to evaporational cooling of the air.

Comparing cloud width on different days can become complicated due to the environmental factors that control cloud size. As is discussed in Hill (1973), the dominant factor governing the size of cumulus clouds is the size and strength of the sub–cloud circulations. There is no way to know what the sub–cloud circulation was for the given days. Only vertical velocity is available, which, as we saw from Panel (a) in Figure 9, was smallest for the in–cloud portions of the L2 flight. Whether the fact that L2 clouds were smaller as compared to the other flights is due to dissipation or environmental characteristics is unknown.

## 5.2 Edge vs. Center Clustering

The finding that clustering is enhanced at smaller spatial/temporal scales agrees with the findings in multiple other papers, including Kostinski and Shaw (2001); Shaw (2003); Shaw et al. (2002); Larsen (2007, 2012). Note that in this paper clustering is measured down to ∼1.8 cm. It is expected from the inertial clustering hypothesis that clustering continues to increase at scales below what was measured here, into the millimeter scales (Shaw, 2003).

PCF curves in other literature (Saw et al., 2008; Shaw, 2003; Shaw et al., 2002) show an elevated value of the PCF at the smallest separations that is naturally accompanied with lower values of the PCF at larger separations. The PCF curves presented here are measured over separations ranging from 1.8 cm (around the Kolmogorov scale) to 12 m (on the order of the integral scale) and show elevated values over a large spatial range (1.8 to 60 cm) before the PCF begins to decay. It is important to keep in mind that this suggests that the elevated PCF values between 12–60 cm are likely to be a result of spatial holes in the droplet concentrations (nonstationary) due to mixing with dryer air, and not preferential concentration from particle inertia.

From the clouds measured, the conclusion can be made that droplet clustering does change as a function of cloud center vs. cloud entrainment (with a large amount of clustering being non–Poissonian, as seen in Table 6), with the entrainment zone having a larger amount of clustering than the center of the cloud, which is shown to be statistically significant. Entrainment is the process by which sub–saturated air surrounding a cloud is drawn into the cloud due to the turbulent motions of the cloudy air, leading to a decrease in the LWC and RH. Cloud top entrainment is evident from looking at Figure 6, Panel (b), which shows the RH and LWC decreasing near cloud top. The entrainment at cloud top can be seen to cause a negative vertical velocity (from evaporative cooling) in the upper portion of the cloud (Figure 6, Panel (c)), where the average vertical velocity is increasingly negative above a normalized cloud height of 0.5, suggesting penetrative downdrafts extend into the middle section of the cloud. Accompanying the cloud top entrainment is a clear increase in the clustering amount.

The production of turbulent energy is equal to the rate of viscous dissipation. Since both production and dissipation depend on the rate of strain $s_{ij}$ (Tennekes and Lumley, 1972), and droplet clustering also depend on $s_{ij}$ from Equation 1, this suggests that droplet clustering depends on turbulence. It was also discussed in Section 2.1 that the Stokes number, which effects the clustering amount, depends on the turbulent dissipation rate. Smith and Jonas (1995) found that the dominant TKE source was at cloud top and was interpreted as evidence for the cloud–top entrainment instability process which produced observed strong downdrafts at cloud top. Shaw (2003) states that one of the main sources of TKE in clouds is from shear evaporative cooling at





cloud edge and cloud top. Kitchen and Caughey (1981) found that turbulent dissipation rates were twice as large at cloud top, which can also be inferred for cloud edge as well.

The findings from previous papers that turbulence is enhanced at cloud top and edge, on top of the fact that entrainment is defined as a mixing process, which in turn is turbulent, could be a possible explanation for the enhanced clustering that is

observed in entrainment and cloud top zones. Unfortunately, the turbulent dissipation rate was not measured in this study due to the lack of 3–D wind data. This work therefore has the potential to be expanded upon by measuring the turbulent dissipation rate and droplet clustering, and determining if there is a strong correlation. Keep in mind that the results presented here provide an overall statistical look at how droplet clustering changes with cloud edge, center, and top, and provides some theories as to how the clustering measured may be related to turbulence. However, the overall numerical values obtained should not be used

quantitatively for precipitation modeling since the data used is non–stationary, and comes from a source (cumulus clouds) that are not known to be stationary.

### 5.3 Aerosol Number Concentration

The results show that the clustering between low and high pollution clouds is statistically similar, suggesting that clustering does not depend on the aerosol number concentration. However, the Stokes number (Equation 2) depends on (1) turbulent

dissipation; (2) droplet size. Even though the droplet sizes between low and high pollution clouds are statistically different, they are still too similar in size (13.4 $\mu$m for low and 10.7 $\mu$m for high) to produce significant differences in $S_t$. For example, assuming the two mean droplet diameters and a turbulent dissipation rate of 100 cm$^2$ s$^{-3}$, a Stokes Number of 0.036 and 0.023 for low and high pollution data, respectively, is obtained. Both Stokes numbers are $\ll 1$, resulting in the droplets following the fluid flow very accurately.

Perhaps the aerosol number concentration can affect the amount of clustering that is occurring if there are significant changes in the sizes of the droplet populations. Such a case could be comparing a highly polluted cloud in Houston (mean droplet diameter $\sim$11 $\mu$m) to a clean cloud over the ocean (mean droplet diameter $\sim$35 $\mu$m, as was found for some Atlantic trade wind cumuli (Wang et al., 2009)). Assuming the Turbulent dissipation rate is constant at 100 cm$^2$ s$^{-3}$, the Stokes number for the polluted cloud would be 0.024, while for the Atlantic trade wind cumuli it would be 0.243, a full order of magnitude greater. A

cloud with a mean droplet diameter of 50 $\mu$m would result in a Stokes number of 0.49. Per Equation 3, a monotonic increase of droplet clustering occurs with increasing Stokes number. Although different authors have come to different conclusions on the exact increase in droplet clustering with different Stokes numbers using direct numerical simulations, it has been found that for a range of $S_t \ll 1$ to $S_t = 0.25$ that the PCF can vary between near homogeneous (zero) amounts of droplet clustering at the lower limit and values between one and two at the upper limit (Chun and Koch, 2005; Wang et al., 2000; Falkovich

and Pumir, 2004). This suggests that the large range of droplet sizes experienced between highly polluted and clean maritime clouds can result in Stokes number differences that produce significant differences in clustering signatures, although this has not been observed/measured in naturally occurring clouds to this point.

Although the conclusion in this paper is that aerosol number concentration does not affect droplet clustering, this conclusion can only be made for the range of N$_a$ given and the resulting mean droplet sizes. From Equation 3, the aerosol number



concentration can affect droplet clustering, given there is a large enough difference in mean droplet size from the aerosol forcing. Also, consider that the droplet size can influence the entrainment rate from increased evaporation of smaller droplets, impacting the turbulent dissipation rate.

## 6 Conclusions

Flight data obtained from the CIRPAS Twin otter aircraft flown during the GoMACCS campaign near Houston, TX from 2006 were used to investigate 81 non–precipitating cumulus clouds, and one vertically developed cumulus cloud, to better understand how droplet clustering changes as a function of cloud location and aerosol number concentration. Of the 22 flights flown, two low (L1, L2) and high (H1, H2) pollution flights were selected to analyze how droplet clustering changed with aerosol number concentration.

It has been shown that (1) droplet clustering is enhanced at cloud edge (and cloud top) as compared to cloud center (statistically significant), and the clustering measured is real, physical variability (non-Poissonian). (2) There is no statistical difference at the 5 percent level for droplet clustering between low and high pollution clouds, at least for the range of $N_a$ that was measured in this research. Although it was found that low pollution clouds do, on average, have a larger amount of clustering in both the center and entrainment zones, this is due entirely to the L2 flight. (3) L1, H1, and H2 have a statistically similar

amount of clustering at the 5 percent level, while L2 has a larger, statistically significant amount of clustering. It is proposed that cloud age plays an important role in the amount of clustering that is occurring, with decaying clouds demonstrating a higher amount of clustering as compared to developing clouds. Although, this theory needs more work as the buoyancy data is not in agreement for decaying clouds.

    This work provides a good statistical base for analyzing how droplet clustering changes with cloud location and aerosol

number concentration, keeping in mind that using the results presented here for quantitative purposes is not wise due to the non–stationarity of the data. The conclusions from this work are drawn only from 81 clouds whose properties are highly variable and influenced by environmental aspects that are not constrained by the observations, including the sub-cloud layer properties and the lifecycle stage of the clouds. Further analysis and data from more clouds is required to confirm some of the ideas that have been presented here. For example, if a field campaign takes place in the future for the purposes of illuminating

these results, constraints on the lifecycle stage of the observed clouds must be considered, along with the proper instrumentation for turbulent analysis.

*Competing interests.* There are no competing interests to declare

*Acknowledgements.* We thank Haflidi Jonsson and Patrick Chuang for their support during the original field campaign. We also thank the CIRPAS Twin Otter crew and personnel for their effort and support during the field program and John Seinfeld and Rick Flagan (Caltech) and



their research groups for assistance and discussions in the field regarding other GoMACCS data sets. Michael Larsen (College of Charleston) also deserves thanks for his suggestions and intellectual help which made this paper possible. This work was funded by NSF CAREERS grant 1255649.



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



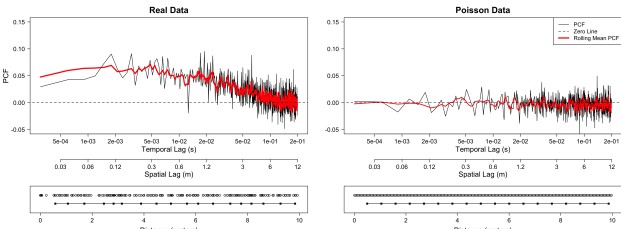

**Figure 1.** The top panels represent the clustering signature for the PCF, with the x–axis showing the temporal and spatial lag on a log–scale and the y–axis representing the PCF (unitless). The top–left shows the PCF for data from a randomly selected portion of cloud covering a temporal range of 2 seconds (∼120 m) with a population of 2168 drops. The top–right panel shows simulated Poisson point data with the same temporal scale and droplet population. The red line represents a rolling mean of five for the PCF values (shown in black). The bottom panels represent short 10 m subsets (∼180 droplets) of the data used in generating the PCF curves. Each circle represents a cloud droplet, and each line with the squares represents the distance traversed for each sampling volume, with each square corresponding to a droplet count of 10.

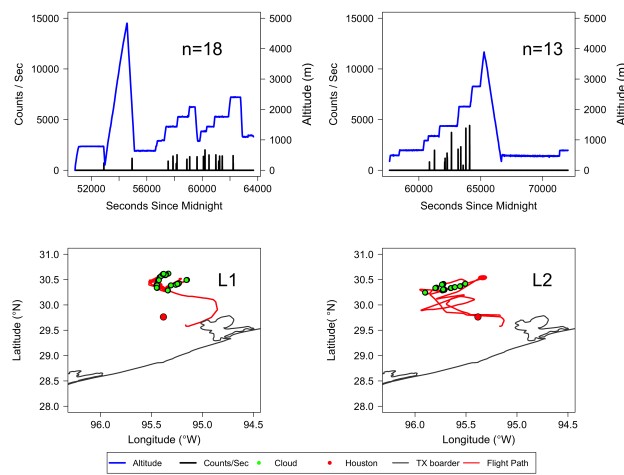

**Figure 2.** Shows L1 and L2 on the left and right, respectively. Flight altitude (blue) as a function of time is displayed in the top panels, with droplet counts per second in black. The "n=" represents the number of clouds sampled for each flight after filtering. The bottom panels show the Texas coast in grey with the location of Houston represented by the red dot. The flight path is outlined in red with the location of clouds displayed by green dots.



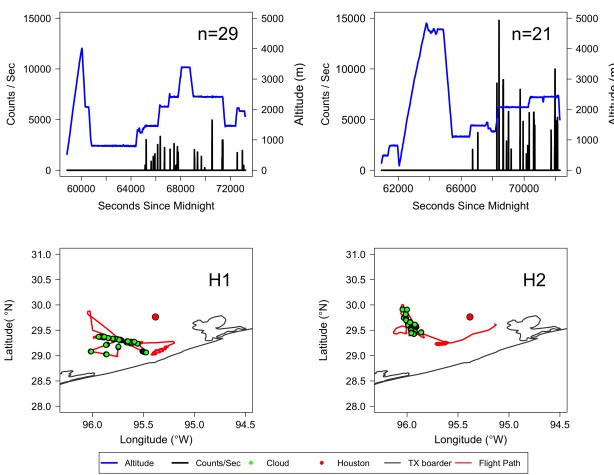

**Figure 3.** As in Figure 1, but for H1 and H2.

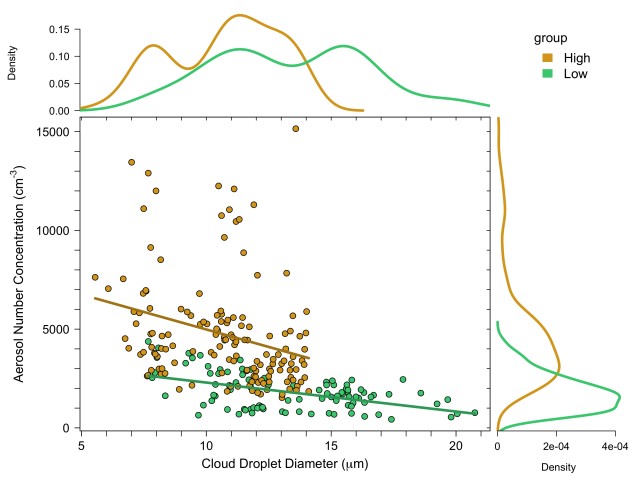

**Figure 4.** Shows cloud droplet diameter ($\mu$m) on the x–axis and aerosol number concentration (cm$^{-3}$) on the y–axis, with low pollution data in green and high pollution data in gold. The corresponding density curves of the high and low pollution data are given on the outer margins of the plot.



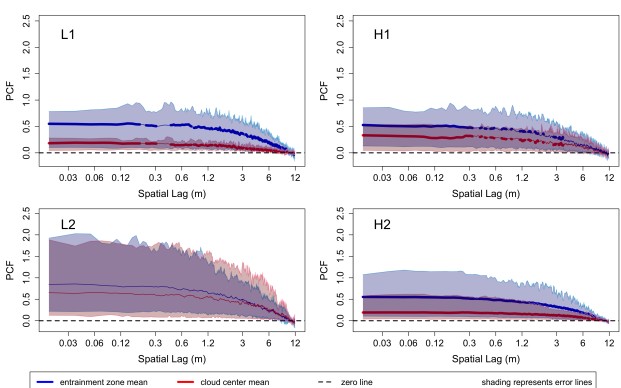

**Figure 5.** PCF clustering signatures for L1 (top right), L2 (bottom left), H1 (top right), and H2 (bottom right) with spatial lag (m) on the x–axis and PCF values (unitless) on the y–axis. Entrainment zone data is in blue and cloud center data is in red, with the envelopes representing the $85^{th}$ (top) and $15^{th}$ (bottom) percent quantile values of the data. The mean PCF value for each case is represented by the middle line in each envelope, where a bold mean line represents entrainment and center differences that are statistically significant.

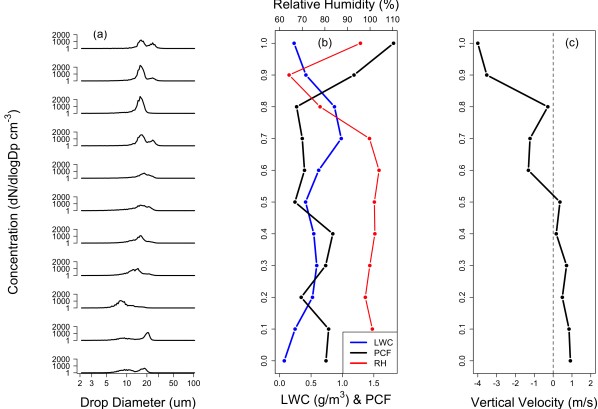

**Figure 6.** Shows the cloud droplet size distribution in panel (a). Panel (b) shows LWC (blue), RH (red), and the PCF (black). Panel (c) shows vertical velocity. All variables are represented as a function of cloud normalized altitude.




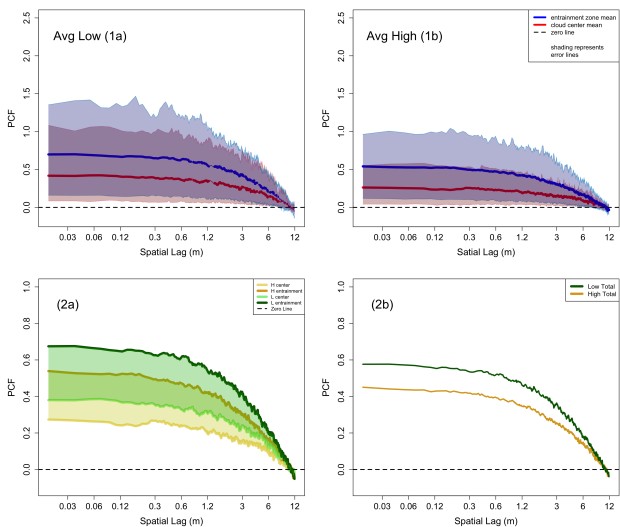

**Figure 7.** Panels (1a) and (1b): as in Figure 5, except for average low pollution PCF values (L1, L2) in Panel (1a) and average high pollution PCF values (H1, H2) in Panel (1b). Panels (2a) and (2b): low pollution data in green and high pollution data in gold. Panel (2a) shows envelopes that span the average center PCF value (lower limit of the envelopes) to the average entrainment PCF value (upper limit of the envelopes). Panel (2b) gives the overall average PCF value for low and high pollution clouds.

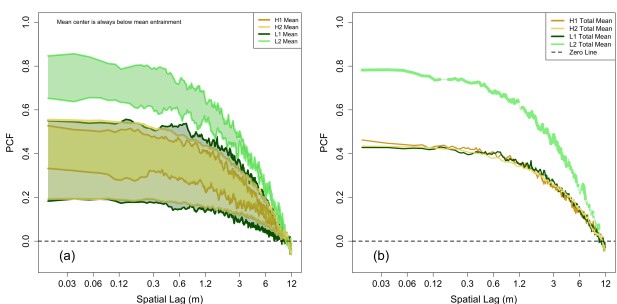

**Figure 8.** As in Panels (2a) and (2b) from Figure 7, except for the individual flights of L1 (light green), L2 (dark green), H1 (light gold), and H2 (dark gold). Note that the envelopes in Panel (a) represents the range from the mean center clustering (lower limit of each envelope) to the mean entrainment clustering (upper limit of each envelope).



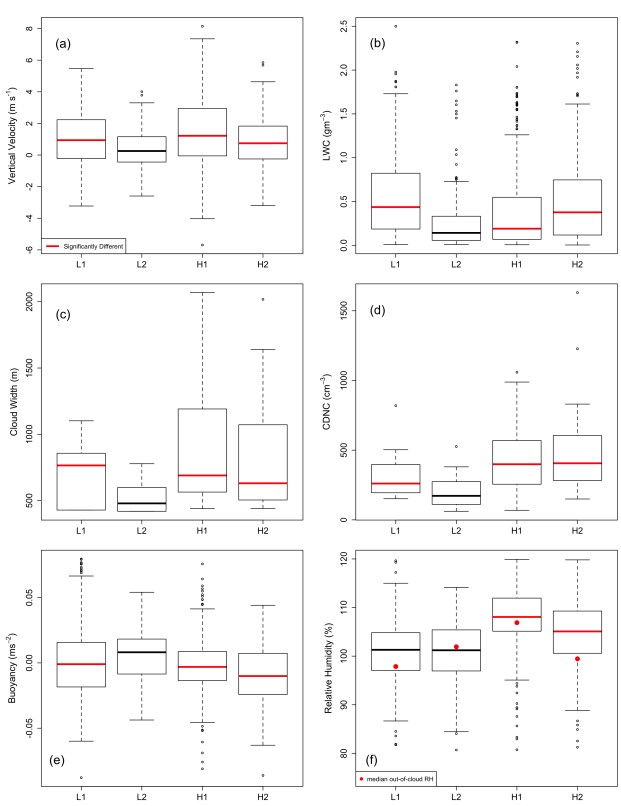

**Figure 9.** Box plots of L1, L2, H1, and H2, represented in that order on the x–axis, with Panels (a) through (f) representing vertical velocity (m s$^{-1}$), LWC (g m$^{-3}$), cloud width (m), CDNC (cm$^{-3}$), Buoyancy (m s$^{-3}$), and in–cloud RH (%), respectively. Red median lines represent datasets that are statistically significant as compared to the L2 dataset. Each red dot in Panel (f) represents out–of–cloud RH.





**Table 1.** Shows the flight information for 20 of the 22 flights that occurred during the GoMACCS campaign. Each flight corresponds to a RF number, date, the number of clouds (after filtering, see text), the total aerosol number concentration ($N_a$), and the accumulation mode aerosol number concentration ($N_{acc}$). Values in parentheses represent the standard deviation.

| Flight | RF Number | Date | Clouds | $N_a$ cm$^{-3}$ | $N_{acc}$ cm$^{-3}$ |
|---|---|---|---|---|---|
| Flight 1 | 1 | 8/21/06 | 1 | NA | NA |
| Flight 2 | 2 | 8/22/06 | 11 | NA | NA |
| Flight 3 | 3 | 8/23/06 | 9 | 2984 (588) | 413 (66) |
| Flight 4 | 4 | 8/25/06 | 17 | 22667 (10672) | 1797 (4184) |
| Flight 5 | 5 | 8/26/06 | 18 | 1409 (762) | 310 (216) |
| Flight 6 | 6 | 8/27/06 | 0 | NA | NA |
| Flight 7 | 7 | 8/28/06 | 0 | NA | NA |
| Flight 8 | 9 | 8/29/06 | 28 | 3157 (1980) | 360 (169) |
| Flight 9 | 11 | 8/31/06 | 37 | 3205 (650) | 972 (214) |
| Flight 10 | 12 | 9/2/06 | 29 | 4768 (2826) | 710 (169 |
| Flight 11 | 13 | 9/3/06 | 0 | NA | NA |
| Flight 12 | 14 | 9/4/06 | 14 | 2770 (758) | 697 (126) |
| Flight 13 | 15 | 9/6/06 | 10 | 1427 (398) | 465 (147) |
| Flight 14 | 16 | 9/7/06 | 32 | 3547 (966) | 949 (306) |
| Flight 15 | 17 | 9/8/06 | 21 | 4824 (1806) | 821 (154) |
| Flight 16 | 18 | 9/10/06 | 1 | 1940 (2125) | 1280 (4172) |
| Flight 17 | 19 | 9/11/06 | 27 | 6561 (5419) | 1280 (4172) |
| Flight 18 | 20 | 9/13/06 | 0 | NA | NA |
| Flight 19 | 21 | 9/14/06 | 25 | 2230 (1157) | 653 (276) |
| Flight 20 | 22 | 9/15/06 | 13 | 2361 (613) | 442 (127) |



**Table 2.** A summary of cloud, flight, and environmental properties from the L1, L2, H1, H2, and Case flights. Note that CDNC stands for cloud droplet number concentration, LWC stands for liquid water content, and Mean Drops ($s^{-1}$) represents the mean number of drops measured by the PDI per second. Standard deviation values are represented in parentheses.

| Variable | L1 | L2 | H1 | H2 | Case |
|---|---|---|---|---|---|
| Date | 2006–Aug–26 | 2006–Sept–15 | 2006–Sept–02 | 2006–Sept–06 | 2006–Sept–10 |
| Flight Number | RF 5–2 | RF 22 | RF 12 | RF 17 | RF 18 |
| UTC for Cloud Sampling | 1447–1717 | 1654–1748 | 1806–2018 | 1832–2002 | 1633–1742 |
| Clouds > 300 m in width | 18 | 13 | 29 | 21 | 1 |
| Min cloud base height (m) | 672 | 1120 | 1457 | 1476 | 806 |
| Max cloud top height (m) | 2412 | 2101 | 2463 | 2451 | 3381 |
| Cloud thickness (m) | 1740 | 981 | 1007 | 976 | 2575 |
| Cloud width (m) | 700 (235) | 520 (110) | 850 (404) | 861 (451) | 943 (472) |
| Mean true air speed (m s$^{-1}$) | 61.2 (1.5) | 59.9 (1.3) | 62.7 (2.3) | 63.0(2.3 | 61.4 (2.2) |
| Mean CDNC (cm$^{-3}$) | 318 (163) | 210 (141) | 421 (255) | 531(363) | 472 (404) |
| Max CDNC (cm$^{-3}$) | 819 | 526 | 1059 | 1630 | 2342 |
| Mean drops (s$^{-1}$) | 661 (448) | 1016 (1037) | 958 (895) | 3300 (2704) | 1679 (1441) |
| Cloud top LWC (g m$^{-3}$) | 0.97 (0.66) | 0.55 (0.54) | 0.47 (0.48) | 0.60 (0.48 | 0.45 (0.35)) |
| Mean vertical velocity (m s$^{-1}$) | 1.81 (1.67) | 1.16 (1.28) | 2.34 (2.21) | 1.61 (1.62) | 0.35 (1.89) |
| $N_a$ (cm$^{-3}$) | 1304 (699) | 2323 (688) | 4331 (2789) | 4650(2057) | 1396 (918) |
| $N_{acc}$ (cm$^{-3}$) | 290 (223) | 436 (159) | 671 (255) | 771 (283) | 943 (472) |
| Environmental lapse rate (°C km$^{-1}$) | 5.4 | 4.5 | 4.8 | 5.7 | NA |
| Environmental RH (%) | 77 | 105 | 74 | 86 | NA |

**Table 3.** Average values for low (L1, L2) and high (H1, H2) pollution clouds for select variables from Table 2

| Variable | Low | High |
|---|---|---|
| Mean CDNC (cm$^{-3}$) | 264 | 476 |
| Mean Drops (s$^{-1}$) | 839 | 2129 |
| $N_a$ (cm$^{-3}$) | 1814 | 4491 |
| $N_{acc}$ (cm$^{-3}$) | 363 | 721 |
| Cloud thickness (m) | 1361 | 992 |
| Cloud width (m) | 610 | 856 |
| Clouds > 300 m in width | 31 (total) | 50 (total) |





**Table 4.** Provides the mean PCF, upper quantile, and lower quantile values for center data (on the left) and entrainment data (on the right) for L1, L2, H1, and H2 in Figure 5, along with average low and high values from Figure 7

| | Center Data | | | Entrainment Data | | |
|---|---|---|---|---|---|---|
| Flight | Mean PCF | Upper Quantile | Lower Quantile | Mean PCF | Upper Quantile | Lower Quantile |
| **L1** | 0.18 | 0.25 | 0.053 | 0.54 | 0.83 | 0.094 |
| **L2** | 0.62 | 1.79 | 0.099 | 0.81 | 1.81 | 0.21 |
| **Avg. Low** | 0.42 | 1.05 | 0.08 | 0.69 | 1.36 | 0.16 |
| **H1** | 0.30 | 0.51 | 0.03 | 0.50 | 0.84 | 0.11 |
| **H2** | 0.19 | 0.58 | 0.04 | 0.54 | 0.098 | 1.13 |
| **Avg. High** | 0.25 | 0.57 | 0.04 | 0.53 | 0.98 | 0.11 |

**Table 5.** Provides the mean p–value and the percent of data that is significant (for the first 21 PCF values) between entrainment and center data for L1, L2, H1, and H2 in Figure 5 and for Average Low and High in Figure 7

| Flight | p–value | % Significant |
|---|---|---|
| **L1** | $6.7 \cdot 10^{-3}$ | 80.9 |
| **L2** | 0.30 | 0 |
| **Avg. Low** | $4.8 \cdot 10^{-3}$ | 100 |
| **H1** | $3.6 \cdot 10^{-3}$ | 90.5 |
| **H2** | $2.3 \cdot 10^{-4}$ | 100 |
| **Avg. High** | $8.2 \cdot 10^{-6}$ | 100 |

**Table 6.** Provides the percentage of clustering that is significant and non–significant for center (C) and entrainment (E) data in L1, L2, H1, and H2.

| Flight | % Significant | % Non–significant |
|---|---|---|
| **L1 C** | 50 | 50 |
| **L1 E** | 55.6 | 44.4 |
| **L2 C** | 84.6 | 15.4 |
| **L2 E** | 80.8 | 19.2 |
| **H1 C** | 55.2 | 44.8 |
| **H1 E** | 60.3 | 39.7 |
| **H2 C** | 85.7 | 14.3 |
| **H2 E** | 95.2 | 4.8 |





**Table 7.** Gives values for vertical velocity (m s$^{-1}$), RH (%), LWC (g m$^{-3}$), the PCF, and the median drop size, respectively, for each normalized cloud height in Figure 6

| Normalized height | Vertical Velocity (m s$^{-1}$) | RH (%) | LWC (g m$^{-3}$) | PCF | Median drop size ($\mu m$) |
|---|---|---|---|---|---|
| 0 | 0.91 | 100.38 | 0.07 | 0.74 | 11.88 |
| 0.1 | 0.84 | 101.01 | 0.24 | 0.78 | 16.43 |
| 0.2 | 0.49 | 97.89 | 0.52 | 0.34 | 8.59 |
| 0.3 | 0.71 | 99.88 | 0.59 | 0.73 | 13.23 |
| 0.4 | 0.16 | 102.12 | 0.54 | 0.84 | 15.28 |
| 0.5 | 0.36 | 101.90 | 0.41 | 0.24 | 15.84 |
| 0.6 | -1.31 | 103.96 | 0.62 | 0.39 | 17.65 |
| 0.7 | -1.22 | 99.74 | 0.98 | 0.36 | 17.03 |
| 0.8 | -0.28 | 77.87 | 0.87 | 0.27 | 15.84 |
| 0.9 | -3.52 | 64.17 | 0.41 | 1.18 | 16.43 |
| 1 | -4.00 | 95.72 | 0.22 | 1.81 | 17.65 |

**Table 8.** Gives the p–value (significant in bold) between every consecutive normalized cloud altitude for PCF values, where (∗) represents a value that is borderline significant/non-significant .

| Comparison | p–value |
|---|---|
| 0 to 0.1 | 0.42 |
| 0.1 to 0.2 | 0.35 |
| 0.2 to 0.3 | 0.46 |
| 0.3 to 0.4 | 0.34 |
| 0.4 to 0.5 | **0.025** |
| 0.5 to 0.6 | 0.26 |
| 0.6 to 0.7 | 0.39 |
| 0.7 to 0.8 | 0.53 |
| 0.8 to 0.9 | **0.066** (∗) |
| 0.9 to 1 | 0.30 |



**Table 9.** Provides the mean p–value and the percent of data that is significant (for the first 21 PCF values) between PCF functions provided in Panel (b) of Figure 8.

| Comparison | p–value | % Significant |
|:---:|:---:|:---:|
| **L2–L1** | $2.9 \cdot 10^{-3}$ | 100 |
| **L2–H1** | $3.3 \cdot 10^{-3}$ | 95.2 |
| **L2–H2** | $7.5 \cdot 10^{-3}$ | 85.7 |
| **L1–H1** | 0.82 | 0 |
| **L1–H2** | 0.92 | 0 |
| **H1–H2** | 0.91 | 0 |

**Table 10.** Median values of vertical velocity (m s$^{-1}$), LWC (g m$^{-3}$), cloud width (m), CDNC (cm$^{-3}$), buoyancy (m s$^{-2}$), in–cloud RH, and out–of–cloud RH (%) from Figure 9.

| Variable Median | **L1** | **L2** | **L1** | **L2** |
|:---:|:---:|:---:|:---:|:---:|
| Vertical velocity (m s$^{-1}$) | 0.94 | 0.25 | 1.21 | 0.74 |
| LWC (g m$^{-3}$) | 0.43 | 0.14 | 0.19 | 0.38 |
| Cloud width (m) | 765 | 480 | 690 | 631 |
| CDNC (cm$^{-3}$) | 261 | 172 | 399 | 406 |
| Buoyancy (m s$^{-3}$) | -0.00097 | 0.0081 | -0.0031 | -0.010 |
| in-cloud RH | 101.3 | 101.2 | 108.1 | 105.1 |
| out–of–cloud RH | 97.8 | 101.9 | 106.8 | 99.4 |

**Table 11.** Gives p–values between L2 and L1, L2 and H1, and L2 and H2 for in–cloud vertical velocity (m s$^{-1}$), LWC (g m$^{-3}$), RH (%), CDNC (cm$^{-3}$), cloud width (m), and in–cloud buoyancy (m s$^{-2}$). Statistically significant values are presented in bold.

| Comparison | Vertical Velocity (m/s) | LWC gm$^{-3}$ | RH (Percent) | CDNC (cm$^{-3}$) | Width (m) | Buoyancy (ms$^{-2}$) |
|:---:|:---:|:---:|:---:|:---:|:---:|:---:|
| L2–L1 | $\mathbf{6.0 \cdot 10^{-4}}$ | $\mathbf{2.1 \cdot 10^{-12}}$ | 0.82 | **0.03** | **0.008** | **0.011** |
| L2–H1 | $\mathbf{2.9 \cdot 10^{-7}}$ | **0.009** | $\mathbf{< 2.2 \cdot 10^{-16}}$ | **0.002** | $\mathbf{7.8 \cdot 10^{-4}}$ | $\mathbf{3.9 \cdot 10^{-5}}$ |
| L2–H2 | **0.011** | $\mathbf{2.2 \cdot 10^{-7}}$ | $\mathbf{3.87 \cdot 10^{-7}}$ | $\mathbf{1.9 \cdot 10^{-4}}$ | **0.003** | $\mathbf{5.5 \cdot 10^{-11}}$ |