# Peer review of "Droplet Inhomogeneity in Shallow Cumuli: The Effects of In–Cloud Location and Aerosol Number Concentration"

_Atmospheric Chemistry and Physics, 2018_

## Referee Comment (RC1) · Anonymous Referee #1 · 1 Oct 2018

Review of "Droplet clustering in shallow cumuli: The effects of in-cloud location and aerosol number concentration" by Dodson and Small-Griswold submitted to Atmospheric Chemistry and Physics (ACP).

Recommendation: reject and encourage resubmission

General evaluation: This paper reports analysis of observations applying a statistical technique that aims at documenting cloud droplet clustering. I found this subject interesting and fitting the ACP scope. However, I am confused by the specific detail of the analysis (PCF normalization) and I feel the way analysis is performed amounts to throwing the baby out with the bathwater. Specifically, forcing the PCF to approach

zero at large scale is not appropriate as there are likely large-scale heterogeneities both at the cloud core (i.e., due to different updraft across the cloud base and thus fluctuations of cloud droplet concentrations above) and at cloud edges due to turbulent mixing and filamentation. I have to admit that I started to read the paper with large expectations, and my enthusiasm went down and down as I kept reading. I admit that I stopped reading at the end of section 4.1. I do feel that the analysis is flawed because of the normalization that forces the cloud to look homogeneous at large scales. There are plenty of cloud observations showing that such an assumption is simply not valid! Thus, I recommend the paper to be rejected and then resubmitted with the discussion based on PCFs without normalization. There are numerous other problems and their sheer number (see specific comments) also suggests the need for a significant rewriting.

Major comments:

1. This comment is arguably more to the ACP technical staff than to the authors. The collection of figures at the end of the manuscript is unacceptable: figures are way to small and the only way to review their details is to go to the electronic version of the paper and zoom in. I feel the journal staff should request the authors to revise the submission before publishing the paper online to have each figure legible (e.g., one figure per page).

2. I do not feel the introduction is appropriate. First, it touches on some issues only remotely related to the specific focus of the paper (e.g., the indirect effects). The selection of references is incomplete or simply inappropriate. Second, the review of previous studies concerning clustering misses important publications as there are studies showing relatively small clustering (in contrast to the papers cited right now). Overall, I agree that there is likely some clustering at the cloud microscale, but its magnitude is relatively small and thus difficult to extract from observations.

3. The discussion in the introduction and in section 2.1 excludes the impact of gravitational acceleration on droplet clustering. This is a serious omission as I would argue that droplet sedimentation is the mean reason for a relatively small droplet clustering. I provide detailed comments below, the Grabowski and Vaillancourt (1999) comment on Shaw et al. (1998) in particular.

4. The way PCF is calculated (section 2.3) need to be better explained. First, I am not sure why normalization is needed. Second, it is not explained how the normalization is performed (a simple shift?). I think there might be interesting differences between cloud cores and cloud edges due to entrainment and small-scale filamentation for the latter (this is what is illustrated in Fig. 1, correct?) In other words, cloud core and cloud edge may look different at different scales. For instance, near the edges, there may be differences at large scales (say tens of centimeters and meters, see Fig. 1), but similar clustering may take place at small-scales (the latter is the focus of the manuscript, correct?). Moreover, normalizing PCF does not allow estimating the magnitude of small-scale concentration fluctuations. The fact that Shaw et al. (2002) did the renormalization is not convincing (although I had just a quick look at that paper without getting into details). I feel this has a significant impact on the results and their interpretation.

Specific comments (those requiring special attention - more serious - marked with *).

1. P. 1, L. 20. I think it would be appropriate to cite Grabowski and Wang (2013) that reviewed the progress in this area a decade after Shaw (2003).

2. P. 1, bottom paragraph: This is a very pessimistic message. I think there are many realistic cloud simulations showing relatively fast rain formation (e.g., vanZanten et al. 2011, Seifert et al. 2010, Wyszogrodzki et al. 2013, Khain et al. 2013).

3. P. 2, L. 5: Earlier references than Khain and Lehman would be appropriate here. For instance, the exchange between Telford and Chai on one side and Jonas and Mason on the other in QJ in 1983 is worth referring to (the great phrase comparing the impacts of entrainment to turning down the gas to boil water faster!). Perhaps some papers of

Charlie Knight at about the same time are also of relevance.

4. P. 2, L. 6: For GN and UGN, there is the Illingworth (1988) paper way before the Knight et al.

5. P. 2, L. 8: "no one theory" is simply incorrect, please see 2 above.

6. P. 2, L. 9-18: this paragraph should be deleted as irrelevant to the results presented. Referring to Small et al. (2009) rather than to original papers (Twomey, Albrecht, etc.) is simply inappropriate. Also, Xue and Feingold (JAS 2006) is a more appropriate first reference to the impact of cloud-edge evaporation impact.

7*. P. 2, L. 19-26: Brenguier and Chaumat (2001) has to be cited here to show that not all studies indicate significant clustering at small scales! In fact, I would argue that the clustering is relatively small (e.g., see Fig. 3 in Kostinski and Shaw 2001 and Fig. 5 in Vaillancourt et al. 2002).

8. P. 2, L. 34: there are many more references than Pinsky pointing out to the significance of droplet clustering for collision/coalescence. Perhaps as reference to a review by Grabowski and Wang (2013) would be appropriate here.

9*. P. 2, L. 35. I do not think the impact of the Reynolds number (i.e., the range of spatial scales) is important for droplet clustering at the microscale. However, the difference in the eddy dissipation rate between laboratory experiments and natural clouds is the key. That was pointed out by Grabowski and Vaillancourt (1999) who also discuss the role of droplet sedimentation and referred to observations that were subsequently reported in Brenguier and Chaumat (2001). Discussion of the latter aspect, droplet sedimentation, is completely missing from the manuscript.

10*. P. 3, last paragraph of the introduction. I think the introduction should lead to the questions posed in this paragraph. The way introduction is written right now does not do that. For instance, why one should expect clustering to depend on the aerosol concentration? Because clustering is expected to depend on the droplet size. Why

it should be different between cloud edge and cloud center? Is that because of the droplet size (arguably smaller at the cloud edge) and intensity of the turbulence (larger at the edges)? Similar for the dependence of the distance from cloud base. Etc. Etc. I feel a complete rewrite of the introduction addressing all points above and leading to these questions is needed.

11. P. 3, L. 23: Kolmogorov scale is around 1 mm in atmospheric turbulence.

12. P. 4. Eq. (1) is valid only for the case without gravity, correct? If so, it is not appropriate for cloud droplets.

13. P. 4, center: the discussion here should include the effects of droplet sedimentation, see Grabowski and Vaillancourt (1999) and perhaps other papers (e.g., from Prof. Lian-Ping Wang?).

14. P. 4, L. 24: the reference to Shaw (2003) is not correct here. The shear near cloud edges comes from cloud dynamics, not evaporation, see original study of Grabowski and Clark (1993) and more recent support from Park et al. (2017).

15. P. 5, L. 1: Should Twomey's or Squiers' old classical papers on droplet activation and growth be referred to here instead of Small and Rosenfeld?

16. P. 5, L. 9: Rather than Pinsky and Khain, the original inhomogeneous mixing papers (John Latham, Marcia Baker, etc.) should be brought here.

17*. P. 6, L. 23. Please explain how the normalization is done. Is the PCF simply shifted up or down to have zero at large scales? If this is an entrainment zone and droplets are clustered at large scales (as clearly illustrated in the left panel of Fig.1), the analysis should show that! This aspect is completely missed by the normalization. If the purpose of the analysis is to show small-scale clustering, then estimating the absolute magnitude of such small-scale clustering is impossible with the normalization.

18*. P. 7, L. 17. Fig. 1 clearly shows that the patchiness is at large scales (meters), not at small scales. Renormalization takes this aspect away. Is the focus of the analysis

on concentration fluctuations at large scale (meters) or at small scales (centimeters) scale? I would think the latter. To me this is the key flaw of the analysis.

19*. Fig. 5. First, I am curious how the figure looks without the normalization (see major point 3 above). The specific discussion in lines 15-20 on p. 9 may change if no normalization is performed. For instance, I think the statement on line 21 ("droplet spacing shifts from non-homogeneous to homogeneous at larger spatial scales") is incorrect as I expect the cloud edge to be quite heterogeneous at large scales (meters and up). The interpretation the authors provide comes from the normalization and it is counterintuitive.

20. Fig. 6. I do not understand what the value of the PCF is shown. Is that the asymptotic values at small scales (i.e., the left edge in Fig. 1)?

As I stated in the overall evaluation, I stopped reading the paper around page 11.

References:

Chaumat, L. and J. Brenguier, 2001: Droplet Spectra Broadening in Cumulus Clouds. Part II: Microscale Droplet Concentration Heterogeneities. J. Atmos. Sci., 58, 642–654.

Grabowski, W. W. and T. L. Clark, 1993: Cloud-environment interface instability, Part II: Extension to three spatial dimensions. J. Atmos. Sci., 50, 555–573.

Grabowski, W. W., and P. Vaillancourt, 1999: Comments on "Preferential concentration of cloud droplets by turbulence: effects on the early evolution of cumulus cloud droplet spectra" by Shaw et al. J. Atmos. Sci., 56, 1433–1436.

Grabowski W. W., and L.-P. Wang, 2013: Growth of cloud droplets in a turbulent environment. Ann. Rev. Fluid Mech., 45, 293-324.

Illingworth, A. I, 1988: The formation of rain in convective clouds. Nature, 336, 754–756.

Khain, A., T. V. Prabha, N. Benmoshe, G. Pandithurai, and M. Ovchinnikov, 2013: The

mechanism of first raindrops formation in deep convective clouds, J. Geophys. Res. Atmos., 118, doi:10.1002/jgrd.50641.

Park, S.-B., T. Heus, and P. Gentine (2017). Role of convective mixing and evaporative cooling in shallow convection. J. Geophys. Res. Atmos. 122, 5351–5363.

Seifert, A., L. Nuijens, and B. Stevens, 2010: Turbulence effects on warm-rain autoconversion in precipitating shallow convections, Q. J. Roy. Meteor. Soc., 136, 1753–1762.

vanZanten, M. C., et al., 2011: Controls on precipitation and cloudiness in simulations of trade‐wind cumulus as observed during RICO, J. Adv. Model. Earth Syst., 3, M06001, doi: 10.1029/2011MS000056.

Wyszogrodzki, A. A., W. W. Grabowski, L.-P. Wang, and O. Ayala, 2013: Turbulent collision-coalescence in maritime shallow convection. Atmos. Chem. Phys., 13, 8471-8487.

Xue, H. and G. Feingold, 2006: Large-Eddy Simulations of Trade Wind Cumuli: Investigation of Aerosol Indirect Effects. J. Atmos. Sci., 63, 1605–1622.
* * *

---

## Referee Comment (RC2) · Anonymous Referee #2 · 15 Oct 2018

The paper is on analysis of field experiment data pertaining to droplet clustering in clouds. This is an important subject considering that the experimental evidence for preferential concentration (inertial clustering) due to interaction with turbulence has been well established in the laboratory since at least a decade ago, yet it has never been satisfactorily documented from field data of atmospheric clouds (as far as I know). The main difficulty stems from statistical non-stationary commonly encountered in cloud data, as the author have discussed in the current manuscript.

In relation to my own criticism, I see it fitting to first clarify that I do not share Referee 1's disapproval (ref: RC1) of the "normalization" done by the authors in the analysis of PCF.

[Figure]

The methodology and the foundation of such practice, the authors failed to disclose, has evolved and, in my view, sufficiently matured in the course of time since Shaw et al. (2002). A mathematical foundation (or origin) of the multiplicative effect of larger scale inhomogeneity on small scale clustering signatures in the Radial Distribution Function (by definition = PCF+1) was exposed (if not proved in strict mathematical sense) in Saw et al. (2012); the same paper also includes a demonstration, using experimental data, of how large scale inhomogeneity resulting from incomplete turbulent mixing caused an apparent uniform upward "shift" of the inertial clustering signatures at small scales. In short, it was shown that the mathematics of how RDF is calculated dictates that: when two concurrent phenomena of inhomogeneities are uncorrelated and occurs with sufficient scale separation, the plainly calculated RDF equals the product of two RDFs, each results from the one of the two phenomena acting alone, i.e. $g(r) = g\_1(r) * g\_2(r)$ .

The above discussion, while relieving the doubts on the normalizing practice, unfortunately triggers another conundrum. Referring to the results (e.g. Figure 4) in Saw et al. (2012), the RDF signature associated with mixing of large (above Kolmogorov) scale inhomogeneity is a curve monotonic increasing with decreasing scale at first, transitioning into a plateau as we approach dissipative scales (the height of the plateau gives a relative measure of the level of inhomogeneity observed). The shape of the corresponding PCF would be qualitatively similar (but somewhat stretched at the large scale end) and this is remarkably similar to all of the PCF seen in the current paper. While, on the other hand, the RDF signature associated with inertial clustering, as found in many previous studies (many direct numerical simulation studies, theories and some experiments e.g. Saw et al. (2012)), have shapes suggestive of power-laws. Thus, it is reasonable to ask if the resulting PCFs in this paper should be more appropriately associated with the inhomogeneity at cloud edges due to entrainment and thus mixing dry and cloudy air, rather than with inertial clustering? Such a view is not inconsistent with the authors own estimate of Stokes numbers of the observed droplets, which is on the order of 0.01 (although we don't really know how good is the 100cmËȩ2/sËȩ3

guess for the energy dissipation rate). Previous works suggests that St âĹij Order(0.01) should results in RDF with power-law exponent of order 0.01 or less, very likely indistinguishable from a plateau or flat horizontal line, given realistic signal-noise ratio of in-situ cloud measurements.

I think major addition/revision is needed to convincingly disentangle mixing signatures from inertial clustering signature, otherwise the paper might have to be rewritten with a different focus (see below).

In relation to points above, perhaps I would suggest, if I might be so bold, that the current data might be better used for careful study of the RDF signatures of the cloud edge entrainment-mixing, which is perhaps also of interest? A good understanding of this will certainly be informative for discovery of proper methods of disentangling entrainment-mixing from inertial clustering in cloud data.

minor comments: page 4, line 31: replace "by" with be page 8, line 1: time should have units in seconds not meter.

Ref: Saw, E. W., Shaw, R. A., Salazar, J. P., & Collins, L. R. (2012). Spatial clustering of polydisperse inertial particles in turbulence: II. Comparing simulation with experiment. New Journal of Physics, 14(10), 105031.

―――――――――――――――――――――――

---

## Author Comment (AC1) · 23 Feb 2019

REPLY: We would like to start by addressing concerns raised by both of the Referees. First, we apologize for the delayed response and posting of the revised manuscript. We put in a great deal of time and effort to do a rewrite of the paper, to shift the focus from the smaller scale clustering originally discussed to larger scale mixing as a result of entrainment. Both Referees raised concerns that the PCFs examined are more closely related to entrainment mixing as compared to preferential concentration, and we have also come to that conclusion. In particular, our PCF spatial scale does not extend into the Kolmogorov range, suggesting that inertial clustering (if present at all) isn't being

measured. Our PCF curves also mirror those presented in Good et al. (2012) and Ireland and Collins (2012), which used the PCF for the purpose of analyzing larger-scale clustering due to entrainment mixing. Lastly, we have de-normalized all PCF curves displayed throughout the manuscript. In Saw et al. (2012) the PCF curves are normalized at the range the larger scale inhomogeneous mixing is occurring (i.e., the 'shoulder' region of the curve). Since the 'shoulder' region of the curve is what we are analyzing, normalizing it to a common value becomes unreasonable.

COMMENT: General evaluation: This paper reports analysis of observations applying a statistical technique that aims at documenting cloud droplet clustering. I found this subject interesting and fitting the ACP scope. However, I am confused by the specific detail of the analysis (PCF normalization) and I feel the way analysis is performed amounts to throwing the baby out with the bathwater. Specifically, forcing the PCF to approach zero at large scale is not appropriate as there are likely large-scale heterogeneities both at the cloud core (i.e., due to different updraft across the cloud base and thus fluctuations of cloud droplet concentrations above) and at cloud edges due to turbulent mixing and filamentation. I have to admit that I started to read the paper with large expectations, and my enthusiasm went down and down as I kept reading. I admit that I stopped reading at the end of section 4.1. I do feel that the analysis is flawed because of the normalization that forces the cloud to look homogeneous at large scales. There are plenty of cloud observations showing that such an assumption is simply not valid! Thus, I recommend the paper to be rejected and then resubmitted with the discussion based on PCFs without normalization. There are numerous other problems and their sheer number (see specific comments) also suggests the need for a significant rewriting.

REPLY: We have realized through your comment that a more clear discussion/explanation of the normalization process was needed. In responses to your other comments that deal with the normalization we will give a more detailed explanation. However, because we have de-normalized all PCF functions in the new manuscript,

normalization no longer applies. To clarify a few things however, we were not attempting to say that there were not large-scale heterogeneities within the cloud. We were simply accounting for said heterogeneity by adjusting all PCF curves to a common value at larger scales (as is done in Shaw et al. (2002), with a much better explanation of the normalization coming in Saw et al. 2012)).

COMMENT: 1. This comment is arguably more to the ACP technical staff than to the authors. The collection of figures at the end of the manuscript is unacceptable: figures are way to small and the only way to review their details is to go to the electronic version of the paper and zoom in. I feel the journal staff should request the authors to revise the submission before publishing the paper online to have each figure legible (e.g., one figure per page).

REPLY: We used the Latex Template that is available through the ACP website. To account for the figures, we adjusted the input parameters to make the figures bigger in the new manuscript. Hopefully this will account for the problem.

COMMENT: 2. I do not feel the introduction is appropriate. First, it touches on some issues only remotely related to the specific focus of the paper (e.g., the indirect effects). The selection of references is incomplete or simply inappropriate. Second, the review of previous studies concerning clustering misses important publications as there are studies showing relatively small clustering (in contrast to the papers cited right now). Overall, I agree that there is likely some clustering at the cloud microscale, but its magnitude is relatively small and thus difficult to extract from observations.

REPLY: We have completely rewritten the introduction by shifting the focus from the small scale inertial clustering to clustering caused by entrainment mixing. We have made an attempt to include and remove references as you recommended in the following comments. We have mentioned that not all studies have measured clustering (see page 3 lines 14-17 in the new manuscript) by citing Chaumat and Brenguier (2001). Although we do not elaborate much, readers will at least come to the realization that

not all studies have measured inhomogeneities.

COMMENT: 3. The discussion in the introduction and in section 2.1 excludes the impact of gravitational acceleration on droplet clustering. This is a serious omission as I would argue that droplet sedimentation is the mean reason for a relatively small droplet clustering. I provide detailed comments below, the Grabowski and Vaillancourt (1999) comment on Shaw et al. (1998) in particular.

REPLY: The original reason we left out a discussion on gravitational acceleration is due to the fact that there is no way to account for it in our observations/data. What we mean is, gravity is a constant in the atmosphere unlike in laboratory experiments where gravity can be accounted for by enhancing or reducing it, as was done in Ireland and Collins (2012). However, you are right that the impact that gravity has is significant, and it is still important for the readers to understand the impacts that gravity has. A discussion on the effects of gravity on droplets has been added on page 5 lines 14-22 of the new manuscript.

COMMENT: 4. The way PCF is calculated (section 2.3) need to be better explained. First, I am not sure why normalization is needed. Second, it is not explained how the normalization is performed (a simple shift?). I think there might be interesting differences between cloud cores and cloud edges due to entrainment and small-scale filamentation for the latter (this is what is illustrated in Fig. 1, correct?) In other words, cloud core and cloud edge may look different at different scales. For instance, near the edges, there may be differences at large scales (say tens of centimeters and meters, see Fig. 1), but similar clustering may take place at small-scales (the latter is the focus of the manuscript, correct?). Moreover, normalizing PCF does not allow estimating the magnitude of small-scale concentration fluctuations. The fact that Shaw et al. (2002) did the renormalization is not convincing (although I had just a quick look at that paper without getting into details). I feel this has a significant impact on the results and their interpretation.

[Figure]

REPLY: You are right that a better job needed to be done to correctly explain the normalization process and why it was needed. I think Referee 2's initial comments do a good job of summarizing the findings in Saw et al. (2012). They found that when two clustering signatures (or inhomogeneities) are occurring and are uncorrelated to one another and have a large enough scale separation (such as inertial clustering on the Kolmogorov scale and entrainment induced clustering occurring on cm to tens of meters scale), the PCF equals the product of two PCFs, each results from the one of the two clustering phenomena acting alone, i.e., $n(t) = n_1(t)*n_2(t)$.

The resulting PCF curve that is produces when both inertial clustering and larger scale inhomogeneities are present is described on page 6 lines 13-21 of the new manuscript. The curve has a power law region at the smallest scales where inertial clustering is present, followed by a 'shoulder' region that is due to the larger scale inhomogeneity. The curve then falls of towards zero at even larger scales (see Figure 4 in Saw et al. 2012). Figure 5 in Saw et al. 2012 goes on to show how the shoulder region of the curve can be 'normalized' to a common value (in this case a value of one for the Radial Distribution Function, in our case it was zero for the PCF). This then allows for analysis of the clustering at the smaller scales.

With that stated, we do believe that the process by which we did the original normalization was flawed. In particular, we were normalizing the PCF curves at the largest scales where the PCF continues to decrease sharply (the region directly after the shoulder region) in order to analyze the differences in the shoulder region. However, in Saw et al. (2012) the PCF is normalized at the shoulder region in order to compare differences in the inertial clustering. We are unaware of any work that has been conducted which shows PCF normalization is valid at the largest scales in order to analyze the shoulder region scale clustering. Although it has been shown that the PCF can be normalized in the shoulder region to analyze inertial clustering, normalizing the PCF at the largest scales to analyze the entrainment scale clustering may or may not be valid.

COMMENT: 1. P. 1, L. 20. I think it would be appropriate to cite Grabowski and Wang

(2013) that reviewed the progress in this area a decade after Shaw (2003).

REPLY: We have kept the original reference of Shaw (2003) but have also added Grabowski and Wang (2013). This section of text now occurs on page 1 line 25 to page 2 line 2.

COMMENT: 2. P. 1, bottom paragraph: This is a very pessimistic message. I think there are many realistic cloud simulations showing relatively fast rain formation (e.g., vanZanten et al. 2011, Seifert et al. 2010, Wyszogrodzki et al. 2013, Khain et al. 2013).

REPLY: The original idea with this was to motivate why droplet clustering is so important, since classical droplet growth theory does such a poor job of estimating the rain formation time. This discussion now appears on page 2 line 4-8. The references you mention have been added on page 2 line 8-11 to state that progress has been made in modeling precipitation. However, we also state that improvements still need to be made, as is mentioned in Wyszogrodzki et al. (2013).

COMMENT: 3. P. 2, L. 5: Earlier references than Khain and Lehman would be appropriate here. For instance, the exchange between Telford and Chai on one side and Jonas and Mason on the other in QJ in 1983 is worth referring to (the great phrase comparing the impacts of entrainment to turning down the gas to boil water faster!). Perhaps some papers of Charlie Knight at about the same time are also of relevance.

REPLY: Entrainment is first mentioned on page 2 lines 11-13 of the new manuscript, and includes the references of Telford and Chai and Jonas and Mason. Since the focus of the paper has been shifted from inertial clustering to clustering caused by entrainment, an extensive discussion on entrainment is given on page 2 lines 22-29 and page 4 lines 13-32 which includes references of some of the earlier work on entrainment, including Warner (1969) on page 2 line 12 and Baker (1980) on page 2 line 24.

COMMENT: 4. P. 2, L. 6: For GN and UGN, there is the Illingworth (1988) paper way

before the Knight et al.

REPLY: The discussion on GN and UGN has been completely removed from the new manuscript.

COMMENT: 5. P. 2, L. 8: "no one theory" is simply incorrect, please see 2 above.

REPLY: We were not trying to imply that none of the theories are correct. We were implying that it is most likely a combination or all of the theories that account for the fast formation of rain that is observed. With that said however, this statement has been removed. And the discussion on the different theories on the fast formation of rain which originally occurred on page 2 lines 3-7 (such as turbulent deviations of supersaturation and GN and UGN) has also been removed to just focus on entrainment mixing and droplet clustering.

COMMENT: 6. P. 2, L. 9-18: this paragraph should be deleted as irrelevant to the results presented. Referring to Small et al. (2009) rather than to original papers (Twomey, Albrecht, etc.) is simply inappropriate. Also, Xue and Feingold (JAS 2006) is a more appropriate first reference to the impact of cloud-edge evaporation impact.

REPLY: This paragraph has been modified in accordance with your comments. We have included Xue and Feingold (2006) which is a modeling experiment, but have also kept Small et al. (2009) since this is an in-situ experiment. Any mention of direct and indirect effects have been removed from the paragraph (which we do agree is irrelevant, especially with how the manuscript has been refocused). However, the mention of the evaporation-entrainment feedback is important, as it explains why we would expect to see enhanced entrainment (and as a result enhanced clustering) for the high pollution clouds as compared to the low pollution clouds. This paragraph can now be found on page 2 lines 16-21 of the new manuscript.

COMMENT: 7*. P. 2, L. 19-26: Brenguier and Chaumat (2001) has to be cited here to show that not all studies indicate significant clustering at small scales! In fact, I would

argue that the clustering is relatively small (e.g., see Fig. 3 in Kostinski and Shaw 2001 and Fig. 5 in Vaillancourt et al. 2002).

REPLY: A brief discussion on the findings of Brenguier and Chaumat (2001) are given on page 3 lines 14-16 of the new manuscript. We also report that not all of the clustering measured shows a statistical difference from a random distribution. In particular see page 9 line 29 to page 10 line 10 of the new manuscript. Also see page 14 lines 1-5, where we state that 31.7 percent of the data we measure (cloud traverses) are randomly distributed.

COMMENT: 8. P. 2, L. 34: there are many more references than Pinsky pointing out to the significance of droplet clustering for collision/coalescence. Perhaps as reference to a review by Grabowski and Wang (2013) would be appropriate here.

REPLY: The quantitative results that were previously discussed have been removed (page 2 lines 30-33 in the original manuscripts) since this is beyond the scope of the current manuscript. It is important that the readers understand how droplet clustering can effect collision coalescence in a conceptual manner however. Page 3 lines 10-13 of the new manuscript have added to Grabowski and Wang (2013) reference in relation to how droplet clustering effects collision-coalescence.

COMMENT: 9*. P. 2, L. 35. I do not think the impact of the Reynolds number (i.e., the range of spatial scales) is important for droplet clustering at the microscale. However, the difference in the eddy dissipation rate between laboratory experiments and natural clouds is the key. That was pointed out by Grabowski and Vaillancourt (1999) who also discuss the role of droplet sedimentation and referred to observations that were subsequently reported in Brenguier and Chaumat (2001). Discussion of the latter aspect, droplet sedimentation, is completely missing from the manuscript.

REPLY: We would agree that the dissipation rate is much more important than the Reynolds number (both for smaller and larger scale clustering), especially since the Stokes number depends on the dissipation rate. The original comment related to the

Reynolds number has been removed. We mention the Reynolds number on page 5 lines 22-26 of the new manuscript, stating that it has been found that large scale clustering does not appear to depend on the Reynold's number as is stated in Ireland and Collins (2012). Note that we still mention the physical mechanisms that small scale clustering depends on. It is important to distinguish the physical differences between small scale inertial clustering and larger scale clustering as a result of entrainment mixing.

COMMENT: 10*. P. 3, last paragraph of the introduction. I think the introduction should lead to the questions posed in this paragraph. The way introduction is written right now does not do that. For instance, why one should expect clustering to depend on the aerosol concentration? Because clustering is expected to depend on the droplet size. Why it should be different between cloud edge and cloud center? Is that because of the droplet size (arguably smaller at the cloud edge) and intensity of the turbulence (larger at the edges)? Similar for the dependence of the distance from cloud base. Etc. Etc. I feel a complete rewrite of the introduction addressing all points above and leading to these questions is needed.

REPLY: We believe the points that you have made here have been met in rewriting the new manuscript. For example, after each of the questions proposed at the end of the introduction, a hypothesis is given which an explanation of why said hypothesis was formed. For example, question 3 on page 3 lines 31-32 states that "it is hypothesized that an increase in aerosol load leads to enhanced clustering due to the resulting increased entrainment from smaller droplet sizes and evaporation". The background information on the reasoning this hypothesis is formed is found on page 2 lines 16-21 of the new manuscript.

COMMENT: 11. P. 3, L. 23: Kolmogorov scale is around 1 mm in atmospheric turbulence.

REPLY: This has been corrected in the new manuscript, as seen on page 2 lines 31-32.

COMMENT: 12. P. 4. Eq. (1) is valid only for the case without gravity, correct? If so, it is not appropriate for cloud droplets.

REPLY: Yes, Eq. (1) was valid only for the case without gravity. This equation has been removed from the revised manuscript.

COMMENT: 13. P. 4, center: the discussion here should include the effects of droplet sedimentation, see Grabowski and Vaillancourt (1999) and perhaps other papers (e.g., from Prof. Lian- Ping Wang?).

REPLY: A discussion on droplet sedimentation has been added as was discussed in the reply to your number 3 major comments, see page 5 lines 15-22.

COMMENT: 14. P. 4, L. 24: the reference to Shaw (2003) is not correct here. The shear near cloud edges comes from cloud dynamics, not evaporation, see original study of Grabowski and Clark (1993) and more recent support from Park et al. (2017).

REPLY: The statement from Shaw (2003) has been removed from the manuscript. Turbulent kinetic energy is no longer discussed in the new manuscript.

COMMENT: 15. P. 5, L. 1: Should Twomey's or Squiers' old classical papers on droplet activation and growth be referred to here instead of Small and Rosenfeld?

REPLY: The statement in which this comment is referring to has been removed from the new manuscript. However, we have cited Twomey (1997) in relation to aerosols and cloud droplet size on page 13 lines 11-12 of the new manuscript.

COMMENT: 16. P. 5, L. 9: Rather than Pinsky and Khain, the original inhomogeneous mixing papers (John Latham, Marcia Baker, etc.) should be brought here.

REPLY: The statement in which this comment is referring to has been removed from the new manuscript. However, since we have shifted the focus to entrainment mixing, we do include multiple citations from Baker, including Baker et al. (1980) on page 2 line 24 and Baker et al. (1984) on page 2 line 32.

COMMENT: 17*. P. 6, L. 23. Please explain how the normalization is done. Is the PCF simply shifted up or down to have zero at large scales? If this is an entrainment zone and droplets are clustered at large scales (as clearly illustrated in the left panel of Fig.1), the analysis should show that! This aspect is completely missed by the normalization. If the purpose of the analysis is to show small-scale clustering, then estimating the absolute magnitude of such small-scale clustering is impossible with the normalization.

REPLY: A detailed description of the normalization process has been provided in the reply to Major comment number 4. But in short, yes the PCF is simply shifted up or down to converge to a set value at larger values (in our case the set value is zero). Again, normalization is no longer done in the new manuscript.

COMMENT: 18*. P. 7, L. 17. Fig. 1 clearly shows that the patchiness is at large scales (meters), not at small scales. Renormalization takes this aspect away. Is the focus of the analysis on concentration fluctuations at large scale (meters) or at small scales (centimeters) scale? I would think the latter. To me this is the key flaw of the analysis.

REPLY: We believe you are correct (along with Referee 2's comments) in stating that the PCFs in this paper should be more appropriately associated with the inhomogeneity occurring at larger scales due to the entrainment of dry air, as opposed to inertial clustering. An explanation of the PCF curve and how it should look for inhomogeneous data is given in the new manuscript on page 6 lines 13-21, and this is also conveyed on page 7 lines 9-12 in describing the PCF for real data calculated in Figure 1. It is believed that the PCF scale we currently use does not extend to small enough scales to even measure inertial clustering (we would need to measure down to mm, whereas we currently only extend down to $\sim$ 3 cm). Lehmann et al. (2007) does not record an increase in the PCF due to inertial clustering for cumuli until length scales of approximately 1mm. We do not extend to the mm scale because we would need to reduce the dt used to calculate the PCF (see page 6 line 24-26), which would result in an even noisier PCF than what is already displayed.

COMMENT: 19*. Fig. 5. First, I am curious how the figure looks without the normalization (see major point 3 above). The specific discussion in lines 15-20 on p. 9 may change if no normalization is performed. For instance, I think the statement on line 21 ("droplet spacing shifts from non-homogeneous to homogeneous at larger spatial scales") is incorrect as I expect the cloud edge to be quite heterogeneous at large scales (meters and up). The interpretation the authors provide comes from the normalization and it is counterintuitive.

REPLY: As can be seen in Figure 5 of the new manuscript, the overall PCF characteristics are unchanged. The main difference in the de-normalized curves is that they no longer converge to zero at the largest spatial lags on the x-axis (some PCF curves are greater than or less than zero). In relation to the discussion that was originally in lines 15-20 on page 9 (which can now be found on page 9 lines 16-22 in the new manuscript), you can see that the overall conclusions remain unchanged. L1, H1, and H2 still remain statistically significant from each other (when comparing center and edge data) while L2 is still statistically similar. In relation to the statement "droplet spacing shifts from non-homogeneous to homogenous at larger spatial scales", this statement was made because we were forcing the PCF to zero (therefore, the statistical significance broke down at larger PCF values), not because homogenous conditions were physically present within the cloud at the time of measurement. Although this statement has been removed in the revised manuscript, this should have been made more clear originally.

COMMENT: 20. Fig. 6. I do not understand what the value of the PCF is shown. Is that the asymptotic values at small scales (i.e., the left edge in Fig. 1)?

REPLY: Yes, the PCF value is the average of the left-most edge of the PCF curve (the first 21 PCF values in the original manuscript). In the revised manuscript, the PCF value shown is an average of the first 60 PCF values that make up the x-axis. In other words, it is the average of the PCF values below $\sim 1$ meter (or the PCF values that make up the 'shoulder' region of the PCF curve).

REFERENCES:

Baker, M., Corbin, R., and Latham, J.: The influence of entrainment on the evolution of cloud droplet spectra: I. A model of inhomogeneous mixing, Quart. J. Roy. Meteor. Soc., 106, 581-598, 1980.

Baker, M., Breidenthal, R., Choularton, T., and Latham, J.: The effects of turbulent mixing in clouds, J. Atmos. Sci., 41, 209-304, 1984.

Good, G., Gerashchenko, S., and Warhaft, Z.: Intermittency and inertial particle entrainment at a turbulent interface: The effect of the large-scale eddies, J. Fluid Mech., 694, 371-398, 2012.

Ireland, P. and Collins, L.: Direct numerical simulation of inertial particle entrainment in a shearless mixing layer, J. Fluid Mech., 704, 301-332, 2012.

Lehmann, K., Siebert, H., Wendisch, M., and Shaw, R.A.: Evidence for inertial clustering in weakly turbulent clouds, Tellus B, 59, 57-65, 2007.

Saw, E. Shaw, R., Salazar, J., and Collins, L.: Spatial clustering of polydisperse inertial particles in turbulence: II. Comparing simulation with experiment, New Journal of Physics, 14, 105031, 2012.
* * *

---

## Author Comment (AC2) · 23 Feb 2019

REPLY: We would like to start by addressing concerns raised by both of the Referees. First, we apologize for the delayed response and posting of the revised manuscript. We put in a great deal of time and effort to do a rewrite of the paper, to shift the focus from the smaller scale clustering originally discussed to larger scale mixing as a result of entrainment. Both Referees raised concerns that the PCFs examined are more closely related to entrainment mixing as compared to preferential concentration, and we have also come to that conclusion. In particular, our PCF spatial scale does not extend into the Kolmogorov range, suggesting that inertial clustering (if present at all) isn't being

measured. Our PCF curves also mirror those presented in Good et al. (2012) and Ireland and Collins (2012), which used the PCF for the purpose of analyzing larger-scale clustering due to entrainment mixing. Lastly, we have de-normalized all PCF curves displayed throughout the manuscript. In Saw et al. (2012) the PCF curves are normalized at the range the larger scale inhomogeneous mixing is occurring (i.e., the 'shoulder' region of the curve). Since the 'shoulder' region of the curve is what we are analyzing, normalizing it to a common value becomes unreasonable.

COMMENT: The paper is on analysis of field experiment data pertaining to droplet clustering in clouds. This is an important subject considering that the experimental evidence for preferential concentration (inertial clustering) due to interaction with turbulence has been well established in the laboratory since at least a decade ago, yet it has never been satisfactorily documented from field data of atmospheric clouds (as far as I know). The main difficulty stems from statistical non-stationary commonly encountered in cloud data, as the author have discussed in the current manuscript.

The methodology and the foundation of such practice, the authors failed to disclose, has evolved and, in my view, sufficiently matured in the course of time since Shaw et al. (2002). A mathematical foundation (or origin) of the multiplicative effect of larger scale inhomogeneity on small scale clustering signatures in the Radial Distribution Function (by definition = PCF+1) was exposed (if not proved in strict mathematical sense) in Saw et al. (2012); the same paper also includes a demonstration, using experimental data, of how large scale inhomogeneity resulting from incomplete turbulent mixing caused an apparent uniform upward "shift" of the inertial clustering signatures at small scales. In short, it was shown that the mathematics of how RDF is calculated dictates that: when two concurrent phenomena of inhomogeneities are uncorrelated and occurs with sufficient scale separation, the plainly calculated RDF equals the product of two RDFs, each results from the one of the two phenomena acting alone, i.e. $g(r) = g\_1(r) * g\_2(r)$.

REPLY: We do believe a better job could have been done in explaining the normalization process due to the confusion that has been shown. Although not discussed or

included in the revised manuscript, originally we believe the process by which we did the normalization was flawed. In particular, we were normalizing the PCF curves at the largest scales where the PCF continues to decrease sharply (the region directly after the shoulder region) in order to analyze the differences in the shoulder region. However, in Saw et al. (2012) the PCF is normalized at the shoulder region in order to compare differences in the inertial clustering. We are unaware of any work that has been conducted which shows PCF normalization is valid at the largest scales in order to analyze the shoulder region scale clustering. Although it has been shown that the PCF can be normalized in the shoulder region to analyze inertial clustering, normalizing the PCF at the largest scales to analyze the entrainment scale clustering may or may not be valid. Although this in and of itself could be an interesting project for future thought. Since we are no longer performing the normalization, we do not mention anything about this within the new manuscript.

COMMENT: The above discussion, while relieving the doubts on the normalizing practice, unfortunately triggers another conundrum. Referring to the results (e.g. Figure 4) in Saw et al. (2012), the RDF signature associated with mixing of large (above Kolmogorov) scale inhomogeneity is a curve monotonic increasing with decreasing scale at first, transitioning into a plateau as we approach dissipative scales (the height of the plateau gives a relative measure of the level of inhomogeneity observed). The shape of the corresponding PCF would be qualitatively similar (but somewhat stretched at the large scale end) and this is remarkably similar to all of the PCF seen in the current paper. While, on the other hand, the RDF signature associated with inertial clustering, as found in many previous studies (many direct numerical simulation studies, theories and some experiments e.g. Saw et al. (2012)), have shapes suggestive of power-laws. Thus, it is reasonable to ask if the resulting PCFs in this paper should be more appropriately associated with the inhomogeneity at cloud edges due to entrainment and thus mixing dry and cloudy air, rather than with inertial clustering? Such a view is not inconsistent with the authors own estimate of Stokes numbers of the observed droplets, which is on the order of 0.01 (although we don't really know how good is the

100cm2 s-3 guess for the energy dissipation rate). Previous works suggests that St Order(0.01) should results in RDF with power-law exponent of order 0.01 or less, very likely indistinguishable from a plateau or flat horizontal line, given realistic signal-noise ratio of in-situ cloud measurements.

REPLY: As already mentioned, we believe you are correct in stating that the PCFs in this paper should be more appropriately associated with the inhomogeneity at cloud edges due to entrainment and the subsequent mixing of dry and cloudy air, as opposed to inertial clustering. An explanation of the PCF curve and how it should look for inhomogeneous data (as you state above) is given in the new manuscript on page 6 lines 13-21, and this is also conveyed on page 7 lines 9-12 in describing the PCF for real data calculated in Figure 1.

We also agree that any inertial clustering that is present would be indistinguishable from the noise the current PCFs display. But it is believed that the PCF scale we currently use does not extend to small enough scales to even measure inertial clustering anyways (we would need to measure down to mm, whereas we currently only extend down to $\sim$ 3 cm). Lehmann et al. (2007) does not record an increase in the PCF due to inertial clustering for cumuli until length scales of approximately 1mm. We do not extend to the mm scale because we would need to reduce the dt used to calculate the PCF (see page 6 line 24-26), which would result in an even noisier PCF than what is already displayed.

COMMENT: I think major addition/revision is needed to convincingly disentangle mixing signatures from inertial clustering signature, otherwise the paper might have to be rewritten with a different focus (see below).

In relation to points above, perhaps I would suggest, if I might be so bold, that the current data might be better used for careful study of the RDF signatures of the cloud edge entrainment-mixing, which is perhaps also of interest? A good understanding of this will certainly be informative for discovery of proper methods of disentangling

entrainment-mixing from inertial clustering in cloud data.

REPLY: As you have suggested, we have completely rewritten the paper to focus on clustering as a result of entrainment mixing as opposed to inertial clustering. We believe that focusing our results on entrainment provides a better analysis and explanation of the PCF signatures that are shown throughout the manuscript.

Although the most significant challenge in all field studies of clouds is distinguishing clustering due to droplet inertia from clustering due to entrainment and mixing (Lehmann et al. 2007), I don't believe the PCF curves displayed here would be appropriate for disentangling entrainment-mixing from inertial clustering since our curves do not extend down into the Kolmogorov range. The fact that our curves don't include inertial clustering also makes them ideal for examining the clustering effect due to entrainment, since we don't have to worry about the Kolmogorov scale effects.

COMMENT: minor comments: page 4, line 31: replace "by" with be page 8, line 1: time should have units in seconds not meter.

REPLY: The section of text that originally included page 4, line 31 has been removed. Units have been changed from meters to seconds (see page 7, line 31).

REFERENCES:

Good, G., Gerashchenko, S., and Warhaft, Z.: Intermittency and inertial particle entrainment at a turbulent interface: The effect of the large-scale eddies, J. Fluid Mech., 694, 371-398, 2012.

Ireland, P. and Collins, L.: Direct numerical simulation of inertial particle entrainment in a shearless mixing layer, J. Fluid Mech., 704, 301-332, 2012.

Lehmann, K., Siebert, H., Wendisch, M., and Shaw, R.A.: Evidence for inertial clustering in weakly turbulent clouds, Tellus B, 59, 57-65, 2007.

Saw, E. Shaw, R., Salazar, J., and Collins, L.: Spatial clustering of polydisperse inertial particles in turbulence: II. Comparing simulation with experiment, New Journal of Physics, 14, 105031, 2012.

---

## Referee Report (RR1)

Review of the revised manuscript "Droplet clustering in shallow cumuli: The effects of in-cloud location and aerosol number concentration" by Dodson and Small-Griswold resubmitted to Atmospheric Chemistry and Physics (ACP).

Recommendation: accept after major revision

General evaluation: This is a resubmission of a manuscript I reviewed previously recommending rejection. I am happy to see that the authors took my comment seriously, modified the analysis, and rewrote the manuscript entirely. In the revised manuscript, the PCFs are no longer adjusted and thus my major objection is no longer valid. Also, the figures are now large enough. The introduction is also completely rewritten and I feel it is more appropriate. Unfortunately, the new manuscript leaves many questions unanswered as detailed below. In general, my comments need to be treated as a review of a new manuscript. I have a few major comments and suggestions, and several minor and more serious technical comments. Overall, I feel the manuscript still requires significant work to put the results on a better footing.

Major comments:

1. One of significant results of the study is the difference between PCFs for the L2 case and the other three cases, the asymptotic value at small scales. I strongly feel that some interpretation of this difference is needed. One possibility would be to expand Fig. 1 so the left panel is replaced with two panels showing PCFs with different asymptotic values similar to the difference between L2 and other cases. Right now, the asymptotic value at small scales in the left panel of Fig. 1 is about an order of magnitude smaller than in Figs. 5, 9 and 10. Why? Should the artificial data be generated in such a way that the PCF has the asymptotic value similar to that in the real data? I simply have no intuition what the asymptotic value means and why there is a difference between L2 and other cases. This needs to be better explained in the revised manuscript.

2. There are numerous unclear statements, phrases, and figure details that make reading and understanding the manuscript difficult. I list several of them in the specific comment section. These needs to be corrected.

3. Section 5.1. This section is a pure speculation that is not supported with any data you present. I really do not know what to suggest. One possibility is to leave this subject out and perhaps replace L2 with another flight that has PCFs similar to L1, H1, and H2. This is also related to 1 above as the manuscript really does not provide any explanation what the difference in the asymptotic value means. I also feel the discussion in section 5.2 (edge versus center) is speculative. Perhaps the discussion in the two sections can be put on a better footing if other cloud parameters are included in the analysis (e.g., adiabatic fraction, spectral width of the droplet size distribution, etc.). At the moment, the PCF analysis alone and the way it is presented in the paper is not sufficient.

4. In view of the above criticism, I am not surprised that the conclusion section 6 is relatively short and thin on merit.

Specific comments (those requiring special attention - more serious - marked with *).

1*. In general, I do not like when the authors refer to concentration inhomogeneities at large scales as "clustering". Yes, the authors make it clear early in the paper that one needs to distinguish between

inertial clustering at sub-cm scales, and concentration heterogeneities at larger scales. However, applying the term "clustering" for the latter in not appropriate in my view. I would prefer to refer to them as "heterogeneities" as they likely result from entrainment and filamentation during turbulent stirring. The authors suggest that these are because of "cloud holes", but I would prefer to think about them as "filaments", not holes. This point is related to my major point 1 above.

2. P2L3 (page 2, line 3): "climate sensitivity" should be explained or a reference that explains this term should be provided.

3. Fig. 1. It is difficult to see that the symbols at the very bottom of the lower graphs are squares. Perhaps replacing squares with crosses or just vertical bars would be better.

4. Beginning of section 2.1. The role of the Stokes number should be brought here. The Stokes number is discussed later, but bringing it in the first paragraph would be more appropriate.

5. P6L2 and 3: "…represents: given…": please rephrase; "time bin": please define.

6. Fig. 5: lower and upper panel heights are different. Please replot.

7*. Figure 6 and its discussion on p. 10 is unclear. Specifically, I do not understand the boxplot, its horizontal scale, and the phrase "that is $\geq$ to the 0.998 quantile". The relation of the boxplot to PCF is unclear. Please explain how one gets the boxplot from the PCF. Or maybe these are different ways to represent the same sequence of droplet positions?

8*. Fig. 7 and its discussion. I think the boxplots are similar to those in Fig. 6, correct? As with Fig. 6, I do not understand what they show. Also, what are the numbers in each panel? The caption does not explain that.

9. Fig. 8. PCF is a function, so I do not understand what Fig 8 shows. Is that the asymptotic value? Also, how come the RH is over 100%? This is also repeated later in the paper (e.g., Table 2 and Fig. 11). This is not possible. Please explain and/or check the data.

10. P11L23: what is the "average PCF value"? Is this different from the asymptotic value I refer to above? Again, I do not know what the difference between the mean values of 0.54 and 0.43 means. And where do they come from (i.e., what does the difference represent)? Again, this goes back to my major comment 1.

---

## Author Response (AR2)

Reply to Comments:

Note that original reviewer comments are in bold below, while the author reply is in regular type font. The tracked changes manuscript is attached after the reply to comments, with removed text in red and added text in blue.

**1. One of significant results of the study is the difference between PCFs for the L2 case and the other three cases, the asymptotic value at small scales. I strongly feel that some interpretation of this difference is needed. One possibility would be to expand Fig. 1 so the left panel is replaced with two panels showing PCFs with different asymptotic values similar to the difference between L2 and other cases. Right now, the asymptotic value at small scales in the left panel of Fig. 1 is about an order of magnitude smaller than in Figs. 5, 9 and 10. Why? Should the artificial data be generated in such a way that the PCF has the asymptotic value similar to that in the real data? I simply have no intuition what the asymptotic value means and why there is a difference between L2 and other cases. This needs to be better explained in the revised manuscript.**

A couple points to comment on here.

(1) The asymptotic value of the PCF is the mean value of the 'shoulder' region of the PCF curve. For example, the asymptotic value of the PCF from the left-hand side of Figure 1 is calculated by taking the mean of the first 60 PCF values (all PCF values that are within the green rectangle in the figure provided below). This provides a way to quantify the PCF curve by giving it a single value. For the case of Figure 1, the asymptotic value is displayed as the thick green line which represents the mean of all the values within the green rectangle). In the revised manuscript, a dashed green line has been added to Figure 1 to give reference to this process, which is described on page 9 lines 15 – 17.

(2) We feel that there may be some confusion here, as the PCF displayed in the left-hand panel of Figure 1 is not synthetic data, but real data collected from a randomly selected cloud. Although the y-axis has a much smaller range in Figure 1 as compared to those of Figures 5, 9 and 10, the figures are in essence showing the same thing. The smaller the PCF, the less inhomogeneous the droplet population is.  The asymptotic value of the PCF displayed on the left-hand side in Figure 1 has a mean value just above 0.05. The bottom of the $15^{th}$ percent quantile envelopes displayed in Figure 5 have mean values of 0.008 (H1 center), 0.090 (H1 edge), 0.022 (L1 center), 0.086 (L1 edge), 0.018 (H2 center), 0.179 (H2 edge), 0.163 (L2 center), and 0.137 (L2 edge). Therefore, the mean PCF value in the left-hand panel of Figure 1 is actually the same magnitude as some of the PCF curves which are embedded in the envelopes that make up Figure 5. Page 7 lines 11-12 of the revised manuscript have added … real data "from a randomly selected cloud" … to hopefully make clear that the PCF from the left-hand panel represents real data.

(3) The asymptotic value is an attempt to give the PCF a single value to define it. The meaning of this value is described on page 6 lines 12-14. For example, it is clear that L2 has a larger average PCF value than that of L1. The mean PCF value for L2 center is 0.61, where the mean PCF value for L1 center is 0.18 (see Table 4). This means that there is a factor of 1.61

enhancement of finding a droplet time t away for L2 center (hear t is the mean time-lag covered in calculating the asymptotic value). There is also a factor of 1.18 enhancement of finding a droplet time t away for L1 center. Therefore, there is more clustering in L2 center as compared to L1 center.

(4) Why there is a difference in L2 as compared to the other three cases is unclear to us as well. We attempt to come up with hypothesis for why there is enhanced inhomogeneities within flight L2 in the discussion section (Section 5.1). However, determining the exact reason for enhanced inhomogeneities in L2 is beyond the scope of this manuscript. We simply set out to determine if (1) inhomogeneities change based on cloud edge or center and (2) if the aerosol number concentration can affect the amount of inhomogeneity. We believe the two goals of this paper have been met with what is presented within the results section (Section 4).

[Figure]

**2. There are numerous unclear statements, phrases, and figure details that make reading and understanding the manuscript difficult. I list several of them in the specific comment section. These needs to be corrected.**

These issues should be corrected in the updated manuscript, please see the reply to specific comments below.

**3. Section 5.1. This section is a pure speculation that is not supported with any data you present. I really do not know what to suggest. One possibility is to leave this subject out and**

**perhaps replace L2 with another flight that has PCFs similar to L1, H1, and H2. This is also related to 1 above as the manuscript really does not provide any explanation what the difference in the asymptotic value means. I also feel the discussion in section 5.2 (edge versus center) is speculative. Perhaps the discussion in the two sections can be put on a better footing if other cloud parameters are included in the analysis (e.g., adiabatic fraction, spectral width of the droplet size distribution, etc.). At the moment, the PCF analysis alone and the way it is presented in the paper is not sufficient.**

Again, a couple points to comment on here.

(1) Section 5.1 was/is part of the discussion section. We are not presenting the data and results displayed in this section as part of the results in the results section, but as a possible hypothesis for why we see the enhanced inhomogeneities in L2. One of your own major comments was the fact that you have no intuition for why there is a difference between L2 and the other three cases. For us to completely ignore L2 and not at least propose ideas/hypothesis for why it is different would be unreasonable as authors. Section 5.1 provided, although not in a definite way, agreed, a possible explanation for why we see the enhanced PCF values for L2. At the very least, Figure 11 shows that flight L2 is statistically different from the other three flights in terms of the vertical velocity, LWC, cloud width, CDNC, and buoyancy. It is our hopes as authors that this discussion will stimulate readers and lead to future work to possibly confirm or deny the hypothesis that we propose. Section 5.1 has been renamed "Cloud Lifetime Hypothesis and Inhomogeneity in L2" as opposed to "Cloud Lifetime Theory" to help clarify that we are proposing a possible hypothesis for the clustering in L2.

(2) Removing flight L2 and replacing it with another flight to make all the flights more familiar in terms of the clustering signatures produced by the PCF would go against the purpose of the paper. We picked flight L2 because it had a sufficient number of clouds and the aerosol number concentration was low. In a perfect world, we would expect larger PCF values for both of the high pollution flights, However, this is not a perfect world and we must account for the results we find, not change the original flights to match a narrative we want or believe we should see.

(3) In section 5.2, we have now added an analysis of the drop size distribution (median drop size and spectral width), along with the cloud droplet number concentration. This tells us with certainty that entrainment is occurring at the edge of the cloud, and the type of entrainment is inhomogeneous (since the drop size distributions are unshifted between edge and center while the cloud drop number concentration is reduced at cloud edge). This added discussion helps in describing why we see enhanced inhomogeneities at cloud edge as compared to cloud center.

**4. In view of the above criticism, I am not surprised that the conclusion section 6 is relatively short and thin on merit.**

Even if the discussion section were longer, the conclusion section we present would still be short. The purpose of the conclusion is to briefly describe the work done and summarize the results. An example of this would be Small et al. (2013), where the conclusion section is a mere 297 words, as compared to the manuscript in question which has 425 words in the conclusion section. The conclusion section, its main purpose, and how it is written varies based on the

authors and their preference.

**1\*. In general, I do not like when the authors refer to concentration inhomogeneities at large scales as "clustering". Yes, the authors make it clear early in the paper that one needs to distinguish between inertial clustering at sub-cm scales, and concentration heterogeneities at larger scales. However, applying the term "clustering" for the latter in not appropriate in my view. I would prefer to refer to them as "heterogeneities" as they likely result from entrainment and filamentation during turbulent stirring. The authors suggest that these are because of "cloud holes", but I would prefer to think about them as "filaments", not holes. This point is related to my major point 1 above.**

We have changes all references to "clustering" that are not related to inertial clustering (small-scale mm-cm clustering) to "inhomogeneities". This can be found throughout the revised manuscript. In particular, the title has been changed to "Droplet Inhomogeneity in Shallow Cumuli: …." Also, see the first paragraph of the revised manuscript on page 1 lines 18 – 23. We have added a discussion describing the cloud 'holes' as areas of decreased droplet concentration due to the entrainment of dry environmental air into the cloud and subsequent filamentation during turbulent mixing. See page 6 lines 20 -24.

**2. P2L3 (page 2, line 3): "climate sensitivity" should be explained or a reference that explains this term should be provided.**

A reference has been added, i.e., Ramaswmy et al., (2001). This reference was in the very first draft of the manuscript, but was mistakenly removed in making the first round of revisions.

**3. Fig. 1. It is difficult to see that the symbols at the very bottom of the lower graphs are squares. Perhaps replacing squares with crosses or just vertical bars would be better.**

The squares have been replaced with vertical bars. We do agree this makes the figure much easier to interpret.

**4. Beginning of section 2.1. The role of the Stokes number should be brought here. The Stokes number is discussed later, but bringing it in the first paragraph would be more appropriate.**

We have added the discussion on the Stokes number directly after the first paragraph in section 2.1. See page 4 lines 10 – 25 of the revised manuscript.

**5. P6L2 and 3: "...represents: given...": please rephrase; "time bin": please define.**

This portion of the text has been rewritten. See page 6 lines 1 – 5. Time bin is defined more precisely in the very next section (Section 2.3). See page 6 lines 26 – 31 for more information on the time bin. After several attempts in trying to define what is meant by time bin when it is first introduced seemed forced or out of place. Particularly when the equation itself (EQ2) provides (albeit indirectly) the time bin when interpreting the meaning of $p(t_o+t \mid t_o)$. We therefore believe

it is more appropriate to directly define the time bin (dt) when discussing how we selected the time bin value in Section 2.3, where we describe the actual process of calculating the PCF.

**6. Fig. 5: lower and upper panel heights are different. Please replot.**

This figure has been corrected. All panels now have the same height.

**7\*. Figure 6 and its discussion on p. 10 is unclear. Specifically, I do not understand the boxplot, its horizontal scale, and the phrase "that is 3 to the 0.998 quantile". The relation of the boxplot to PCF is unclear. Please explain how one gets the boxplot from the PCF. Or maybe these are different ways to represent the same sequence of droplet positions?**

Your last statement here is correct. Figure 6 is simply a different way to represent the same sequence of droplet positions. The histogram data represented in figure 6 is of the mean inter-arrival times which are used in calculating the PCF. In the simplest terms, we are looking at the distribution of inter-arrival times before the data has been plugged into the PCF. We have added the text on page 10 lines 20 -22 stating "We can use the inter-arrival times used in calculating the PCF to develop a better understanding of the overall droplet spatial distributions and the largest inhomogeneities measured by directly analyzing the inter-arrival time distribution". This statement directly tells the reader what it is they are looking at.

The boxplot is taking the largest 0.2 percent of data represented in the histograms (i.e., all values greater than or equal to the 0.998 quantile of the data). This is a more precise way of analyzing how the largest spatial inhomogeneities between droplet measurements are distributed between edge and center data. The horizontal scale of the boxplot comes directly from the x-axis of the histogram (since the boxplot is taking the largest data points from the histogram). We have changed the $\geq$ symbol with "greater than or equal to" on page 10 line16 of the revised manuscript, along with adding "i.e., the largest 0.2 percent of the IAD data) on page 10 lines 17 of the revised manuscript in order to clarify any misunderstanding.

Hopefully this response clarifies that he boxplot is coming directly from the histogram data. And it is the histogram data that is used to create the PCF curves. Therefore, one does not get the boxplots from the PCF.

**8\*. Fig. 7 and its discussion. I think the boxplots are similar to those in Fig. 6, correct? As with Fig. 6, I do not understand what they show. Also, what are the numbers in each panel? The caption does not explain that.**

Yes, the boxplots for figure 7 are derived from the same histogram shown for the L1 case, but from the histograms from the L2, H1, an H2 cases (not shown). The left-hand panel of Figure 7 (L1) is the exact same boxplot as is shown in Figure 6. It is included so a comparison can be made between the other 3 cases. This fact has been added to the caption. An explanation of the numbers in each panel have also been added to the figure caption (that explanation is already contained in the text on page 10 line 33 of the revised manuscript.

**9. Fig. 8. PCF is a function, so I do not understand what Fig 8 shows. Is that the asymptotic**

**value? Also, how come the RH is over 100%? This is also repeated later in the paper (e.g., Table 2 and Fig. 11). This is not possible. Please explain and/or check the data.**

Yes, Figure 8 shows the asymptotic value of the PCF. We have made this more clear by stating "…Shows how the mean PCF value (where again, the mean PCF value is obtained by taking the mean of all individual PCF values to the left of the green reference line in Figure 1) …" on page 11 lines 3 – 4 of the revised manuscript. The revised manuscript has corrected all RH values. We have filtered the RH to remove bad data (i.e., data that is over 103 %). The discussion on filtering RH values can be found on page 8 lines 14 -15 of the revised manuscript. Note that this issue has also been discussed in other papers working GoMACCS RH data, including Small et al., (2013) and Lu et al., (2008).

**10. P11L23: what is the "average PCF value"? Is this different from the asymptotic value I refer to above? Again, I do not know what the difference between the mean values of 0.54 and 0.43 means. And where do they come from (i.e., what does the difference represent)? Again, this goes back to my major comment 1.**

We believe that this comments has been addressed in previous replies. The "average PCF value" has been changed to "mean PCF value" on page 11 line 34 to try and keep consistency throughout the manuscript. But yes, the mean PCF value or the "average PCF value" as originally states refers to the asymptotic value. The mean value of 0.54 means there is a factor of 1.54 times more clustering as compared to a homogeneous distribution. The mean value of 0.43 means that there is a factor of 1.43 times more clustering as compared to a homogeneous distribution. The difference therefore represents that there is more inhomogeneity in the droplet population with a mean value of 0.54 as compared to the inhomogeneity in the droplet population with a mean value of 0.43.

REFERENCES:

[revised manuscript text omitted]

---

## Author Response (AR3)

Reply to Comments:

Note that original reviewer comments are in bold below, while the author reply is in regular type font. The tracked changes manuscript is attached after the reply to comments, with removed text in red and added text in blue.

**1. P4L1 (page 4, line 1): This IS a first…?**

The word "is" has been added to this sentence. Please see page 4, line 1 of the revised manuscript.

**2. P7L4: What do you mean by "seen above"?**

The phrase "seen above" has been changed to "discussed in the previous paragraph. Please see page 7, line 4 in the revised manuscript.

**3. P8L8: "that occurred" is not needed.**

"Not occurred has been removed. Please see page 8 line 8 of the revised manuscript.

**4. P12L1 and 4: "Although it appears" is applied twice. Perhaps one can be changed?**

The second sentence has been rewritten to remove the "although it appears". Please see page 12 line 6 of the revised manuscript.

**5. P13L6: a reference to studies discussing the subsiding shell would be desirable here (e.g., Heus and Jonker JAS 2008).**

Several new references have been added in relation to the humid shell. Please see page 13 lines 8 and 9 of the revised manuscript.

**6. P14L2: Hill was published in 1974. I expect there are more recent papers discussing this aspect.**

An additional reference was added to accommodate this comment. Please see page 14 line 2 of the revised manuscript.

[revised manuscript text omitted]